# How spatial release from masking may fail to function in a highly directional auditory system

Norman Lee*†, Andrew C Mason

Department of Biological Sciences, Integrative Behaviour and Neuroscience Group, University of Toronto Scarborough, Toronto, Canada

**Abstract** Spatial release from masking (SRM) occurs when spatial separation between a signal and masker decreases masked thresholds. The mechanically-coupled ears of *Ormia ochracea* are specialized for hyperacute directional hearing, but the possible role of SRM, or whether such specializations exhibit limitations for sound source segregation, is unknown. We recorded phonotaxis to a cricket song masked by band-limited noise. With a masker, response thresholds increased and localization was diverted away from the signal and masker. Increased separation from 6° to 90° did not decrease response thresholds or improve localization accuracy, thus SRM does not operate in this range of spatial separations. Tympanal vibrations and auditory nerve responses reveal that localization errors were consistent with changes in peripheral coding of signal location and flies localized towards the ear with better signal detection. Our results demonstrate that, in a mechanically coupled auditory system, specialization for directional hearing does not contribute to source segregation.

*For correspondence: lee33@stolaf.edu

Present address: †Department of Biology, St. Olaf College, Northfield, Minnesota, United States

Competing interests: The authors declare that no competing interests exist.

## Introduction

Natural soundscapes are composed of numerous biotic and abiotic sound sources (*Brumm and Slabbekoorn, 2005*; *Theunissen and Elie, 2014*). Even in the presence of natural noise, auditory systems are adept at detecting, recognizing, and localizing sound sources of interest for animals that rely on hearing to mediate appropriate behavioural decisions (*Bee and Micheyl, 2008*; *Lee et al., 2017*). Segregating sounds of interest from noise can be challenging because sound waves from multiple sources combine to form a composite waveform prior to auditory transduction. From this input, coherent representations of various sources are derived from the composite waveform and signals of interest are segregated from other competing sound sources (*McDermott, 2009*). In humans, this sensory challenge is known as the 'cocktail party problem' and it describes difficulties encountered in perceiving speech in noisy social settings (*Bregman, 1990*; *Bronkhorst, 2000*; *Brumm and Slabbekoorn, 2005*; *Cherry, 1953*; *Hulse, 2002*; *McDermott, 2009*).

Non-human animals may encounter analogous situations in natural acoustic environments (*Bee, 2012*; *Bee and Micheyl, 2008*; *Hulse, 2002*; *Römer, 2013*). Some insect choruses are known to produce sustained levels of background noise (*Gerhardt and Huber, 2002*; *Römer, 2013*). These choruses are often composed of mixed species with communication signals that may share spectral, temporal, and spatial overlap. Such background noises have the potential to negatively impact the performance of auditory systems in perceiving relevant signals (*Brumm, 2013*; *Brumm and Slabbekoorn, 2005*; *Wiley, 2015*).

Hearing with two ears provides important computational advantages for sound localization and sound source segregation (*Schnupp and Carr, 2009*). When two ears are spatially separated, each will experience different sound arrival times and sound pressure levels in response to an incident

sound wave. Interaural time and level differences (ITD and ILD respectively) are binaural cues that the auditory system can utilize to compute sound direction (*Knudsen and Konishi, 1979*; *Michelsen, 1998*). Spatially separated sound sources will give rise to different ITDs and ILDs for each source and sound diffraction between the two ears can generate different signal-to-noise ratios associated with each source leading to a 'better ear advantage' for detecting a particular sound source (*Blauert, 1997*). Both these auditory cues may contribute to spatial release from masking (SRM), a sensory mechanism that allows for improved reception of target signals that are spatially separated from noise sources (*Bee and Micheyl, 2008*).

Spatial release from masking has been documented in ears that function as independent pressure receivers and in ears that are internally (acoustically) coupled to function as pressure difference receivers (*Bee, 2007*; *Bee and Christensen-Dalsgaard, 2016*; *Brunnhofer et al., 2016*; *Caldwell et al., 2016*; *Dent et al., 1997*; *Hine et al., 1994*; *Nityananda and Bee, 2012*; *Saberi et al., 1991*; *Schwartz and Gerhardt, 1989*; *Sümer et al., 2009*; *Ward et al., 2013*). Whether spatial release from masking may contribute to sound source segregation in mechanically coupled pressure receivers remains unknown. Mechanically coupled ears have been described in some tachinid and sarcophagid flies that rely on directional hearing to localize the advertisement signals of suitable host species (*Allen, 1995*; *Cade, 1975*; *Lakes-Harlan et al., 1999*; *Robert et al., 1999*). Upon source localization, larvae are deposited and develop as parasites within hosts (*Adamo et al., 1995*; *Allen et al., 1999*; *Cade, 1975*).

The ability of *Ormia ochracea* (Diptera: Tachinidae) to localize cricket calling songs is remarkable because their small physical size severely limits conventional acoustic cues for sound localization (*Michelsen, 1998*; *Robert, 2005*). With bilaterally symmetric ears that are separated by a mere 500 µm (*Robert et al., 1996a*), diffraction of the ~7 cm wavelength field cricket calling song is insufficient to result in measurable acoustic ILDs and the best possible ITD is 1.45 µs for a sound source at 90° relative to the midline (*Robert et al., 1996b*). Despite these minuscule sound field directional cues, *O. ochracea* are still capable of localizing sound sources to the accuracy of 2° azimuth (*Mason et al., 2001*). This hyperacute directionality is made possible by the mechanical coupling of the two tympana with a flexible cuticular lever that amplifies the sound field ITDs into larger mechanical ITD and ILDs sufficient for neural processing (*Oshinsky and Hoy, 2002*). The surprising acuity of directional hearing in *O. ochracea* has led to considerable interest in mechanically coupled auditory systems as models for novel acoustic technology (e.g. *Bauer et al., 2016*; *Kuntzman and Hall, 2014*), as well as other applications involving localization of signal sources subject to the so-called 'aperture problem' (e.g. *Akçakaya and Nehorai, 2010*).

In addition to computing sound direction, *O. ochracea* must also recognize the calling songs of preferred host crickets (*Gray et al., 2007*). Field cricket calling songs consist of tonal sound pulses with a dominant frequency of about 5 kHz that are repeated at a species-specific pulse rate (*Walker, 1962*). After song detection, *O. ochracea* engage in flying (*Müller and Robert, 2001*) and/or walking phonotaxis to the source (*Lee et al., 2009*; *Mason et al., 2005*). Success in localizing host crickets requires the auditory system to integrate two types of temporal information. The auditory system must measure a difference in the timing of sound pulses that arrive at the two ears to determine sound direction, and the timing between consecutive sound pulses for song pattern recognition (species recognition). In this study, we examine whether spatial release from masking may contribute to improved host song recognition and localization in the presence of masking noise. We exploit the directional characteristics of the mechanically coupled tympana and use a point-source band-limited noise to asymmetrically interfere with the temporal pattern input to both sides of the auditory system. If spatial release from masking confers hearing advantages in *O. ochracea*, behavioural response thresholds are expected to decrease while localization accuracy is expected to increase when noise is spatially separated from a signal of interest (relative to noise and signal sources in close proximity). However, over a range of spatial separations between a target signal and masker (from 6° to 90° of spatial separation), we find no support for SRM in auditory responses of *O. ochracea*. Rather, a localized masker interferes with auditory directionality, and this effect is more severe for a larger spatial separation between the target signal and masker.

## Results

In behavioural experiments, we used a spherical treadmill system (*Lott et al., 2007*; *Mason et al., 2001*) to measure walking phonotaxis to a frontal target signal (attractive synthetic cricket trill) and a laterally positioned masker (band-limited noise) (*Figure 1A*). Test stimuli consisted of a 76 dB SPL 4 s burst of band-limited masker in combination with a 2 s target signal broadcast at different intensities to result in varied signal-to-noise ratios (SNRs) (*Figure 1B*). Test stimulus presentation started with the broadcast of the masker. The target signal was presented 1 s after the onset of the masker such that the target signal was embedded in the masker.

### Walking responses to the target signal and masker in isolation

Flies remained quiescent in the absence of stimulus presentation. When presented with the standard 76 dB SPL 2 s cricket song from 0° azimuth (*Figure 1A*) flies responded with a latency of $56.45 \pm 2.76$ ms after stimulus onset; forward velocity increased to a sustained level of $6.03 \pm 0.28$ cm/s (*Figure 2A*) while steering velocity fluctuated around 0 cm/s (*Figure 2B*). Flies exhibited sustained walking with an angular heading of $2.93 \pm 5.25°$ and covered a total distance of $16.91 \pm 0.91$ cm (*Figure 2C*). When subjected to the masker alone from the forward location (0° azimuth) at the same intensity, flies responded with a similar latency of $52.19 \pm 3.32$ ms (Paired Sample T-Test: n = 13 flies t = 0.741, p = 0.477). After masker onset, forward velocity increased transiently to levels less than those observed in response to the target signal, but rapidly declined to near baseline levels (*Figure 2A*). Steering velocity appeared to fluctuate more sporadically around 0 cm/s (*Figure 2B*) as flies responded with a different mean angular heading of $0.76 \pm 4.03°$ (Watson $U^2$ Test: n = 13, $U^2 = 0.256$, p = 0.01). Although walking responses were directed toward the masker location, responses were short-lived and thus flies traveled a significantly shorter total distance of $2.08 \pm 1.01$ cm (Paired Sample T-Test: n = 13 flies, t = 9.983, p < 0.001, *Figure 2C*). Flies did not perform negative phonotaxis (away from the masker) in response to the masker in isolation.

### Increased spatial separation does not decrease masked behavioural response thresholds

We estimated behavioural response thresholds using an approached described in *Bee and Schwartz (2009)*. Behavioural response thresholds depended on the masking condition (Repeated Measures ANOVA: $F_{(1.12, 12.33)} = 72.36$, p < 0.001, *Figure 3*). In quiet, flies responded with phonotaxis to the target signal at a mean threshold intensity of $50.80 \pm 5.44$ dB SPL. Adding a masker that was spatially separated by 6° from the frontal signal (small separation) significantly increased response thresholds to $63.31 \pm 2.64$ dB SPL ($F_{(1, 11)} = 66.39$, p < 0.001). Increasing the spatial separation from 6° to 90° (large separation) between the target signal and masker resulted in a mean response threshold of $63.19 \pm 2.96$ dB SPL that did not differ significantly from response thresholds when the signal and masker was separated by 6° ($F_{(1, 11)} = 0.07$, p = 0.80).

### Effects of SNR and source separation on sound localization

Response latencies for the target signal were not significantly affected by SNR (Repeated Measures ANOVA: $F_{(2,12)} = 0.412$, p = 0.671) or source separation ($F_{(1,6)} = 0.001$, p = 0.974). Flies responded with an average latency of $59.51 \pm 3.51$ ms. Total distance travelled did not depend on source separation ($F_{(1,9)} = 0.092$, p = 0.768) but was significantly affected by SNR ($F_{(2,18)} = 8.601$, p = 0.002). A linear contrast revealed that walking distance increased with SNR (linear contrast: $F_{(1,9)} = 15.456$, p = 0.03). With an increase in SNR from −6 dB to +6 dB, walking distance increased from $11.35 \pm 1.81$ cm to $16.36 \pm 1.12$ cm. Post Hoc contrasts revealed that a SNR of −6 dB resulted in significantly shorter distances walked than at higher SNRs (−6 dB vs 0 dB SNR: $F_{(1, 1)} = 8.466$, p = 0.017; −6 dB vs +6 dB SNR: $F_{(1, 1)} = 15.456$, p = 0.003).

Forward velocity profiles reveal a strong but brief walking component in response to the masker that quickly decayed before song onset (*Figure 4A*). During song presentation, forward velocity increased as a function of SNR (Repeated Measures ANOVA: $F_{(2,14)} = 25.381$, p < 0.001) but not song and masker separation ($F_{(1,7)} = 1.866$, p = 0.214). With an increase in SNR from −6 dB to +6 dB SNR, average forward velocity increased from $2.26 \pm 0.46$ cm/s to $4.67 \pm 0.39$ cm/s but all were significantly less than the average forward velocity of $6.03 \pm 0.28$ cm/s in response to the song in isolation (Control vs. −6 dB SNR: $F_{(1,9)} = 56.86$, p < 0.001, Control vs. 0 dB SNR: $F_{(1,9)} = 46.79$,

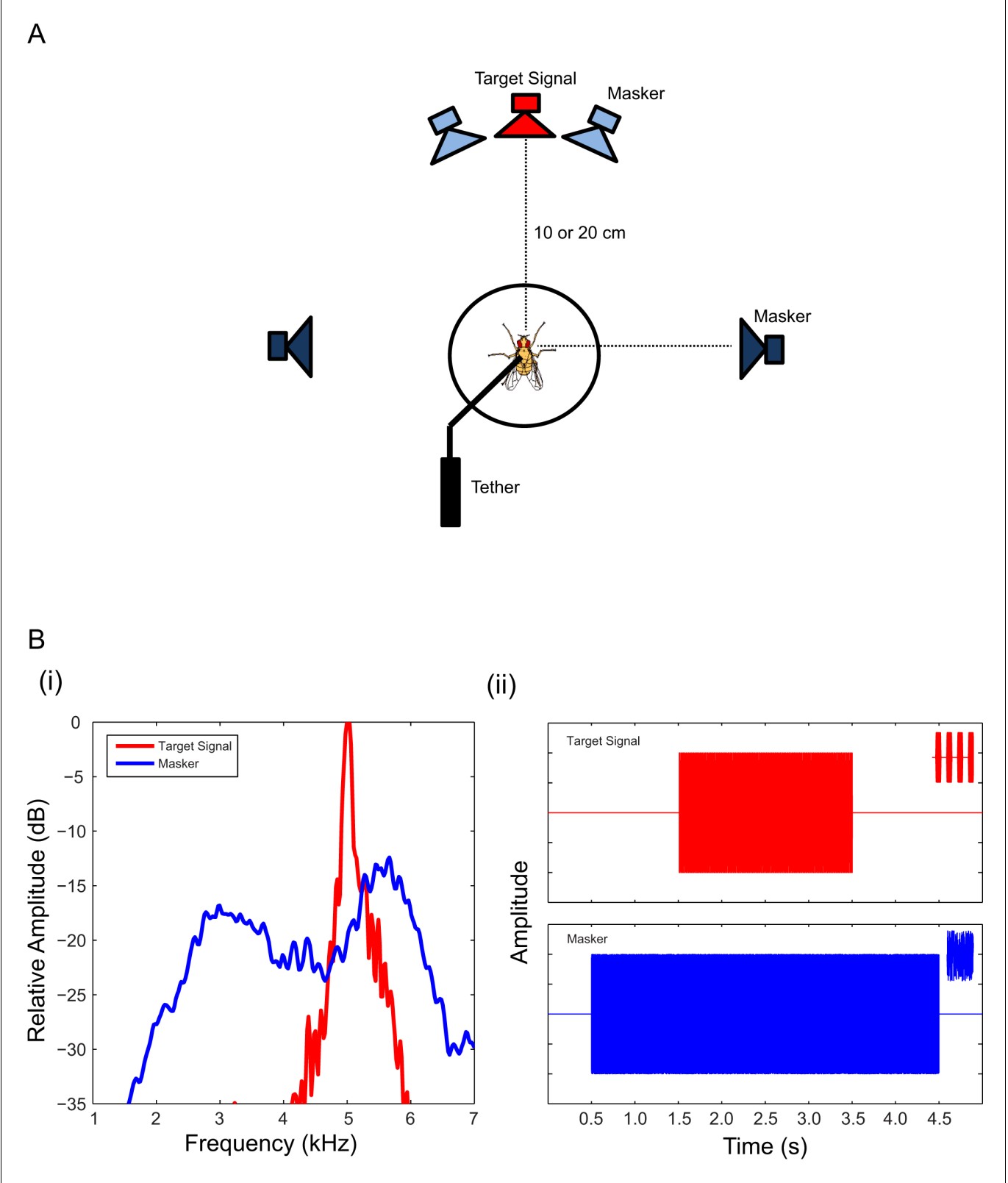

**Figure 1.** Signal and masker and experimental setup used in recording walking phonotaxis. (A) Gravid female *Ormia ochracea* tethered on top of the treadmill system and positioned equidistant (Experiments 1 and 3: 20 cm, Experiment 2: 10 cm) from surrounding speakers. The target signal (synthetic cricket song) was broadcast from the forward (red) speaker and the masker (band-limited noise) was broadcast from an adjacent speaker (light blue – either to the left or right of the target signal speaker) separated by 6°, or from a laterally positioned speaker (dark blue - either to the left or right of the

*Figure 1 continued*

target signal speaker) separated by 90°. (**B**) (i) Spectral analysis of the target signal (ii - top panel) and masker (ii – lower panel). Insets in (ii) show expanded samples of stimulus waveforms. In test conditions, the masker (4 s in duration) was broadcast 0.5 s post-trigger for data acquisition, followed by the simultaneous broadcast of the target signal after a delay of 1.5 s.

$p < 0.001$, Control vs.+6 dB SNR: $F_{(1,9)} = 26.24$, $p = 0.001$). SNR-dependent changes in forward velocity corresponded with SNR-dependent changes in walking distances (above).

At the onset of the masker, flies initiated small, transient steering responses toward the masker (negative velocities) with durations that depended on masker intensity ($F_{(2,14)} = 7.804$, $p = 0.05$, *Figure 4B*) but not location ($F_{(1,7)} = 1.828$, $p = 0.218$, *Figure 4B*). Steering toward the masker lasted between $65.00 \pm 20.44$ ms and $157.01 \pm 29.59$ ms as masker intensity was dropped from 82 dB to 70 dB SPL. This was followed by a quick transition to steering away from the masker.

Overall walking direction (angular heading) depended on the spatial separation between the signal and masker (Two-way Circular ANOVA (hk test): n = 13, $F_{(1,65)} = 8.0324$, $p = 0.0061$, *Figure 4C*), but not SNR ($F_{(2,65)} = 1.2105$, $p = 0.3047$), or the interaction between source separation and SNR ($F_{(2,65)} = 0.7298$, $p = 0.4859$). In response to the simultaneous broadcast of the target signal and masker from speakers separated by 6° (small separation), there was a slight tendency for flies to deviate further from the control direction with greater directional variability at the lowest SNRs. With a short latency after the onset of signal presentation, flies steered away from the masker with an overall average velocity of $0.37 \pm 0.15$ cm/s that did not significantly change to different SNRs ($F_{(2,14)} = 1.115$, $p = 0.355$, *Figure 4B*). Flies responded with similar angular headings (−6 dB: $−10.19 \pm 14.99°$, 0 dB: $−4.85 \pm 7.41°$, +6 dB: $−5.69 \pm 4.41°$) for all SNR treatments and these differed significantly from controls ($2.93 \pm 5.25°$) (Hotelling's Pair-Sample Test: control vs. −6 dB SNR, n = 11, F = 5.7276, $p = 0.0249$; control vs. equal dB SNR, n = 13, F = 7.0532, $p = 0.0107$, control vs. +6 dB SNR, n = 13, F = 14.1137, $p = 0.0012$, *Figure 4C*).

In response to the simultaneous broadcast of target signal and masker separated by 90° (large separation), flies oriented to different mean angular headings at different SNRs (−6 dB: $−15.02 \pm 7.42°$, 0 dB: $−12.75 \pm 3.75°$, +6 dB: $−10.39 \pm 2.19°$), all of which were significantly different from responses to controls (control vs. −6 dB SNR, n = 13 F = 6.6895, $p = 0.0238$; control vs. equal dB SNR, n = 13, F = 37.8136, $p < 0.0001$, control vs. +6 dB SNR, n = 13, F = 43.6432, $p < 0.0001$, *Figure 4C*). At −6 dB SNR, angular heading did not differ for the two source separations (−6 dB SNR 6° vs. 90°: F = 0.7372, n = 13, $p = 0.5173$). However, signal and masker separation caused a significant shift contralateral to the location of the masker for 0 dB and +6 dB SNR respectively (0 dB SNR 6° vs. 90° separation: F = 7.0648, n = 13, $p = 0.0106$, +6 dB SNR 6° vs. 90° separation: F = 3.9304, n = 13, $p = 0.055$, *Figure 4C*). Steering velocity depended on source separation ($F_{(1,7)} = 9.193$, $p = 0.019$, *Figure 4B*). When the signal and masker were separated by 90°, steering velocity increased to $1.07 \pm 0.34$ cm/s and movement was directed away from the target signal and masker (*Figure 4C*).

## Effects of balanced temporal pattern interference on localization accuracy

Additional behavioural experiments were conducted to establish whether binaurally balanced noise would restore accurate signal localization. In these experiments, we equalized temporal pattern interference with a second coherent (identical) masker positioned at a location that is a direct mirror reflection of the other masker location (*Figure 1A*). When the signal was presented in isolation, flies walked a total distance of $7.74 \pm 0.45$ cm and localized a frontal 76 dB SPL signal with an angular heading of $−0.32 \pm 3.1°$ (*Figure 5A*). When the masker was broadcast along with the target signal at equal intensity and spatially separated by 6° to the right (*Figure 5B*) or left (*Figure 5C*) of the target signal speaker, flies responded with mean angular headings contralateral to the masker location (right masker: $−13.46 \pm 7.95°$, left masker: $11.46 \pm 7.80°$). When the signal and masker were separated by 90° (*Figure 5E,F*) flies responded with significantly different mean angular headings (6° vs. 90° separation right masker: n = 12, $U^2 = 0.51$, $p < 0.001$, 6° vs. 90° separation left masker: n = 12, $U^2 = 0.51$, $p < 0.001$) that were diverted further away from the signal location (right masker: $−47.04 \pm 11.91°$, left masker: $46.93 \pm 16.08°$). With a second identical masker, regardless of whether

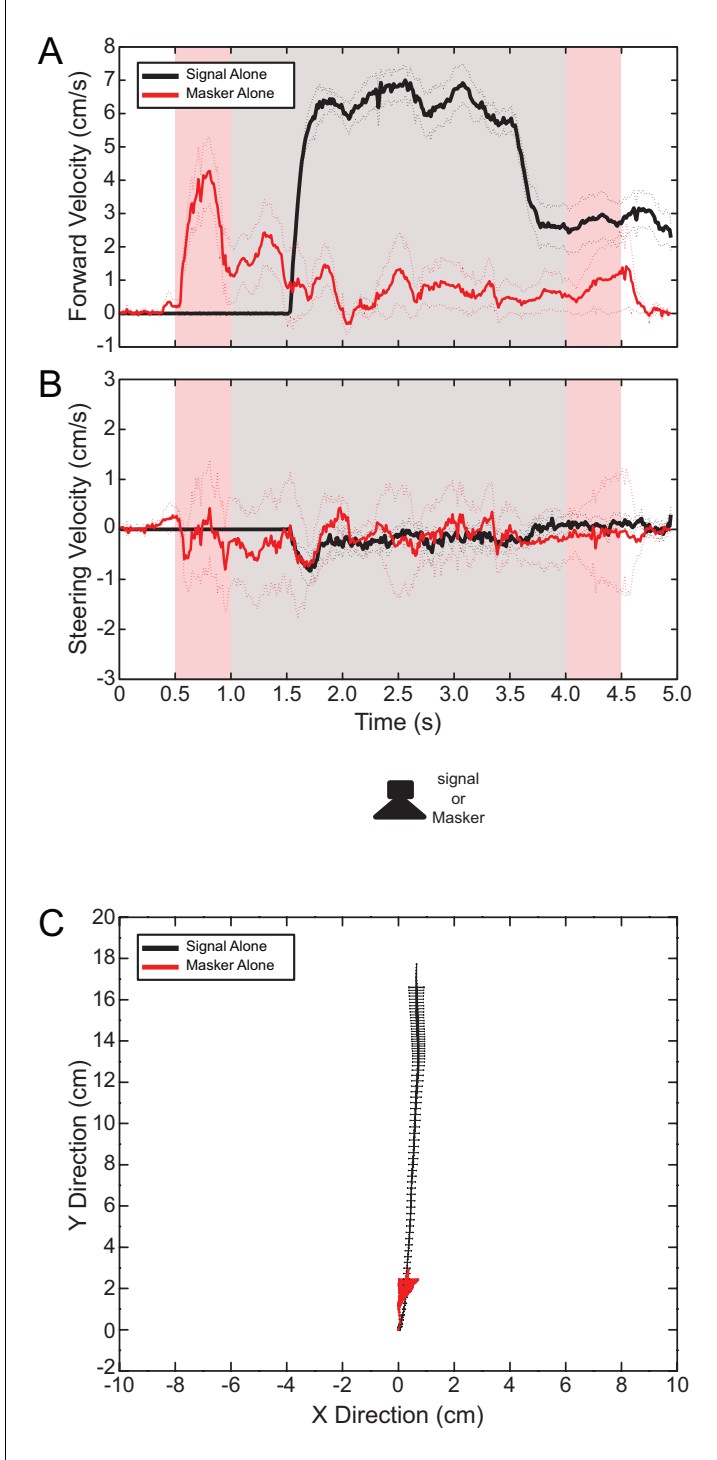

**Figure 2.** Walking phonotaxis in response to the target signal and masker in isolation. Mean (solid lines) and SEM (dotted lines) of (**A**) forward and (**B**) steering velocity components from walking phonotaxis in response to a forward (0°azimuth) target signal (black) or masker (red) presented in isolation. Light pink shaded areas indicate the duration of masker presentation. Gray-shaded areas indicate the duration of signal presentation. (**C**) Virtual walking trajectory showing responses directed to the target signal (black) or masker (red). Walking responses were robust and accurately directed to the target signal. In contrast, flies only walked transiently in response to the masker presented alone.
The following source data is available for figure 2:

*Figure 2 continued on next page*

*Figure 2 continued*

**Source data 1.** Translational velocity in response to the target signal and masker presented in isolation.
**Source data 2.** Steering velocity in response to the target signal and masker presented in isolation.
**Source data 3.** Virtual walking path in response to the target signal and masker presented in isolation.

the signal and masker were separated by 6° (*Figure 5D*) or 90° (*Figure 5G*), flies consistently walked less ($F_{(1,11)}$ = 4.209, p = 0.04) and localized a forward direction (6° separation: 1.93 ± 3.35°, 90° separation: 2.82 ± 6.19°) that did not differ from responses to a forward target signal in quiet (signal alone vs. maskers with a 6° separation: n = 12, $U^2$ = 0.05, p > 0.5, signal alone vs. maskers with a 90° separation: n = 12, $U^2$ = 0.12, p > 0.1).

## Tympanal measurements

We hypothesized that these results could be explained by the interaction of an asymmetrically placed masker with the directional properties of *O. ochracea's* peripheral auditory system, which would differentially bias the perceived SNR at each ear. Because the masker-ipsilateral ear would show a stronger masker-driven response than the contralateral, whereas the signal-driven responses of the two ears would be equal. We refer to this ear-specific SNR as the *effective amplitude* of

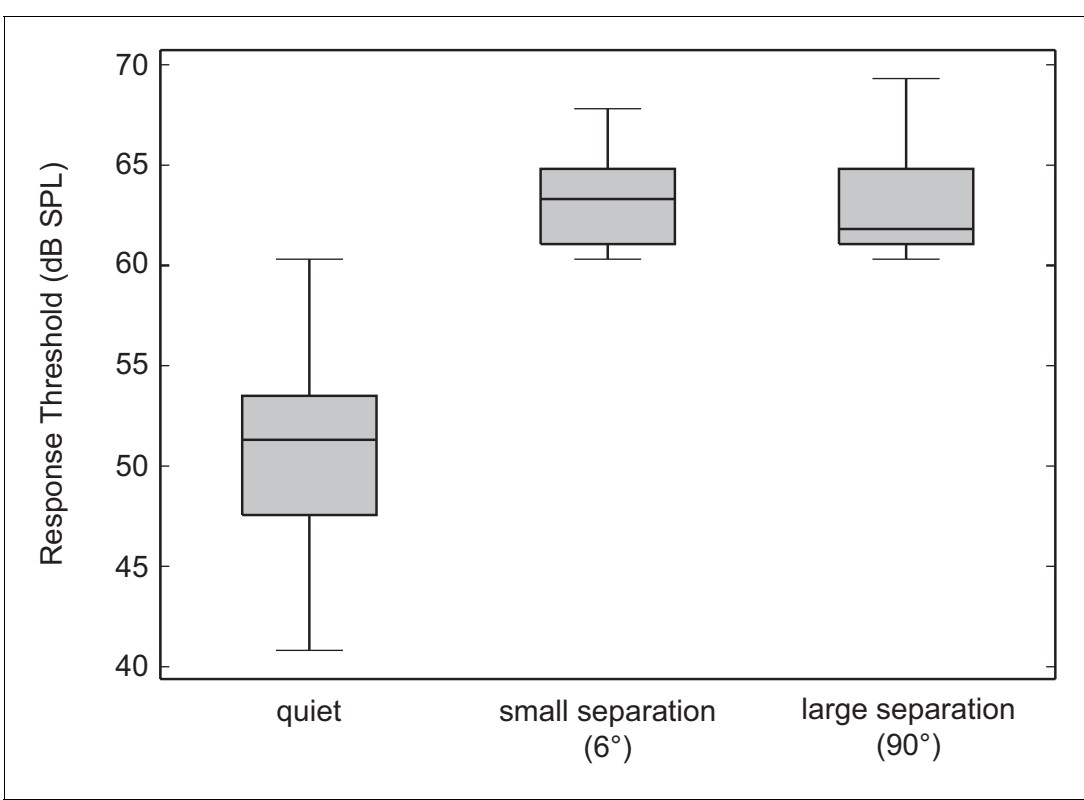

**Figure 3.** Spatial separation between target signal and masker does not decrease behavioural response thresholds. Estimated behavioural response thresholds in quiet, with a target signal and masker separated by 6°, or 90°. Box plots depict first, second (median), and third quartiles. Whiskers depict 1.5 × interquartile range. The presence of the masker resulted in an increase in behavioural response thresholds compared to the target signal in quiet. Increased spatial separation between the target signal and masker does not improve response thresholds.
The following source data is available for figure 3:

**Source data 1.** Behavioral response thresholds to a target signal in quiet, and signal and masker separated by 6° or 90°.

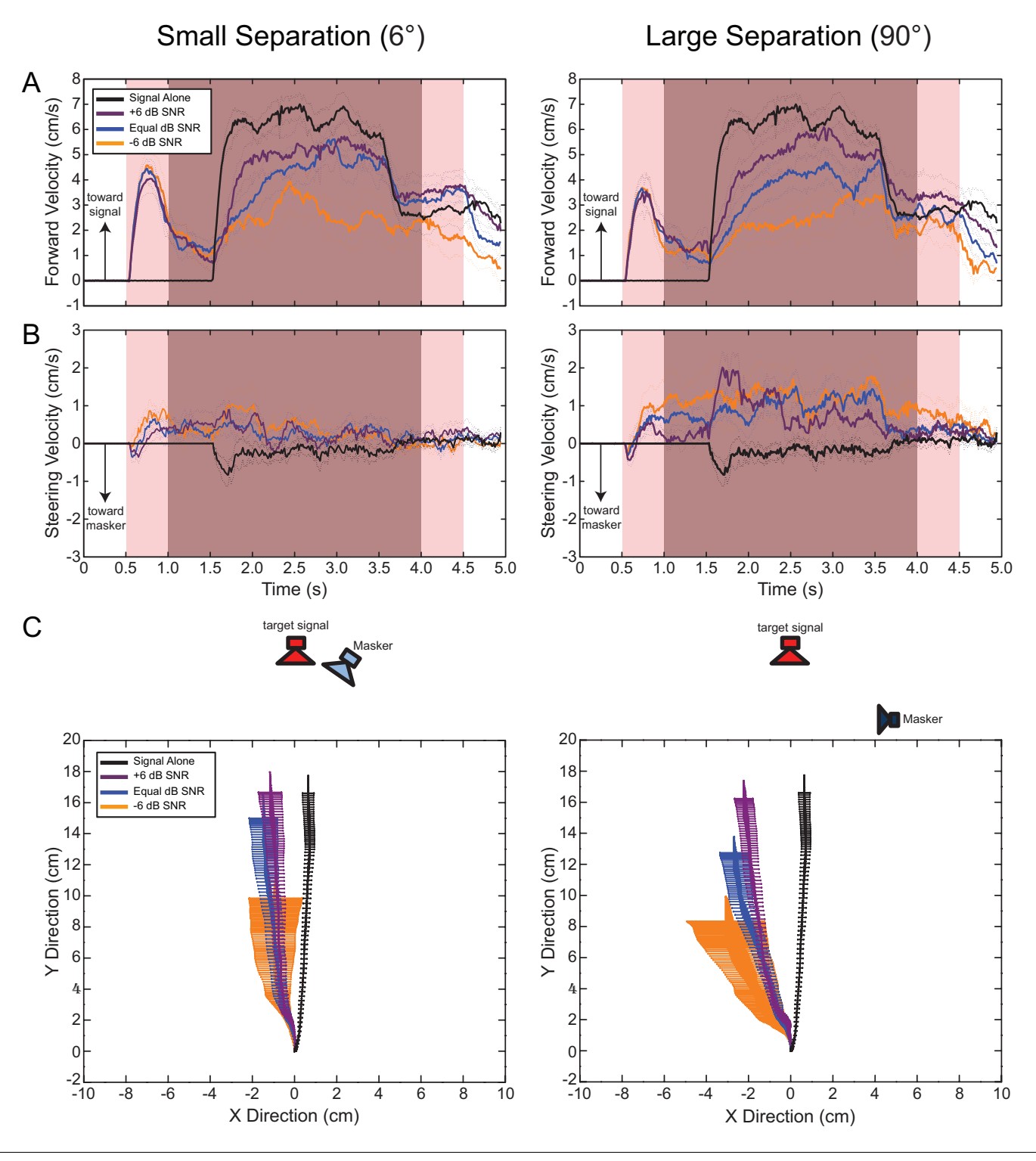

**Figure 4.** Effects of spatial separation and signal-to-noise ratio on sound localization. Mean (solid lines) and SEM (dotted lines) of (**A**) forward and (**B**) steering velocity components to a 76 dB SPL target signal alone (black trace) or signal and masker broadcasted at SNRs of −6 dB (orange trace), equal dB (blue trace), +6 dB (purple trace) from signal and masker with a 6° (left) or 90° (right) spatial separation. Light pink shaded areas indicate the duration of masker presentation. Dark pink shaded areas indicate the duration of the simultaneous presentation of signal and masker. At masker onset, flies responded with a brief walking component in the forward direction that quickly decayed. During this time frame, flies reactively steered to the masker, followed by steering movements away from the masker that were maintained for the remainder of stimulus presentation. Steering responses were minimally affected by SNR but were enhanced at greater signal and masker separation. At song onset, forward velocities increased to speeds well

*Figure 4 continued on next page*

*Figure 4 continued*

above responses to the masker onset. Average forward velocities depended on SNR but not signal and masker separation. Flies maintained steering away from the masker location. (**C**) Virtual walking trajectories in response to the target signal and masker separated by a 6° (left) or 90° (right) separation. Walking distance changed with SNR but not with signal and masker separation. The masker caused walking responses to be diverted away from the locations of the target signal and masker. With a 90° spatial separation walking responses were diverted even further in a direction contralateral to the masker location.

The following source data is available for figure 4:

**Source data 1.** Translational velocity in response to a target signal and masker separated by 6° and presented at different signal-to-noise ratios (SNRs).
**Source data 2.** Steering velocity in response to a target signal and masker separated by 6° and presented at different signal-to-noise ratios (SNRs).
**Source data 3.** Virtual walking path in response to a target signal and masker separated by 6° and presented at different signal-to-noise ratios (SNRs).
**Source data 4.** Translational velocity in response to a target signal and masker separated by 90° and presented at different signal-to-noise ratios (SNRs).
**Source data 5.** Steering velocity in response to a target signal and masker separated by 90° and presented at different signal-to-noise ratios (SNRs).
**Source data 6.** Virtual walking path in response to target signal and masker separated by 90° and presented at different signal-to-noise ratios (SNRs).

responses (i.e. response to target signal above the response to masker) to avoid confusion with the SNR of the broadcast stimulus, which we manipulated in our experiments (*Figure 6A*). The overall effect of such a bias in effective response amplitude would be to mimic binaural cues for a lateral signal source location (Figure 9). We verified that this was the case by recording tympanal vibrations with a laser Doppler vibrometer (LDV). We recorded tympanal responses for repeated presentations of a target signal and masker separated by 6° or 90°, matching speaker configurations in behavioural experiments (*Figure 6*). Both tympana vibrate at the onset of the masker and continued to do so throughout masker presentation. Vibration amplitudes were elevated during target signal presentation. When the signal and masker were separated by 6°, the effective amplitude was only slightly larger in the masker-contralateral ear compared to the masker-ipsilateral ear (*Figure 6B,D*). When the spatial separation between the target signal and masker was increased to 90°, there was a pronounced increase in effective amplitude measured from the masker-contralateral ear (*Figure 6C,D*).

To quantify this effect, tympanal vibrations were recorded in response to a range of broadcast SNRs when the target signal and masker were separated by 90°. Effective response amplitude depended on the SNR (Repeated Measures ANOVA: n = 6, $F_{(1.08, 5.41)}$ = 34.13, p = 0.001) and masker location ($F_{(1, 5)}$ = 14.44, p = 0.013), but not the interaction between SNR and masker location ($F_{(1.10, 5.46)}$ = 2.46, p = 0.173). The effective amplitude increased with increasing SNR and was greater for the masker-contralateral tympanum across all SNRs (*Figure 7A*).

We determined the 'effective' interaural vibration amplitude difference (IVAD) as a difference in the effective response amplitude between the ears. We applied our IVAD measurements and previously published tympanal directionality data (*Robert et al., 1996b*) to generate predictions of the apparent incident sound direction for the target signal embedded in the masker (*Figure 7B*). Predicted incident sound directions closely matched mean angular headings measured in behavioural experiments across all SNRs (*Figure 7C*). For example, at equal SNR (0 dB), the interaural effective amplitude difference was measured to be −2.76 ± 0.54 dB which translates to a predicted incident sound direction of −15.10 ± 5.16°. At this SNR, flies localized a sound direction of −12.75 ± 3.75° and this did not differ significantly from the predicted incident sound direction of the target signal (Watson-Williams F-test: F = 0.17, p = 0.69).

A SNR of −6 dB resulted in an interaural effective amplitude difference of 1.71 ± 0.75 dB and a predicted incident sound direction of 10.85 ± 6.79°. A mean interaural effective amplitude of 2.76 ± 0.54 dB was measured for a SNR of 0 dB and this corresponds to an incident sound direction of 15.10 ± 5.16°. When the SNR was increased to +6 dB, this resulted in a measured interaural effective amplitude difference of 2.35 ± 0.66 dB and a predicted sound direction of 13.48 ± 6.39°. These apparent incident sound directions, predicted by effective response amplitude differences, closely matched actual angular headings in behavioural responses (see above) (all p-values > 0.05).

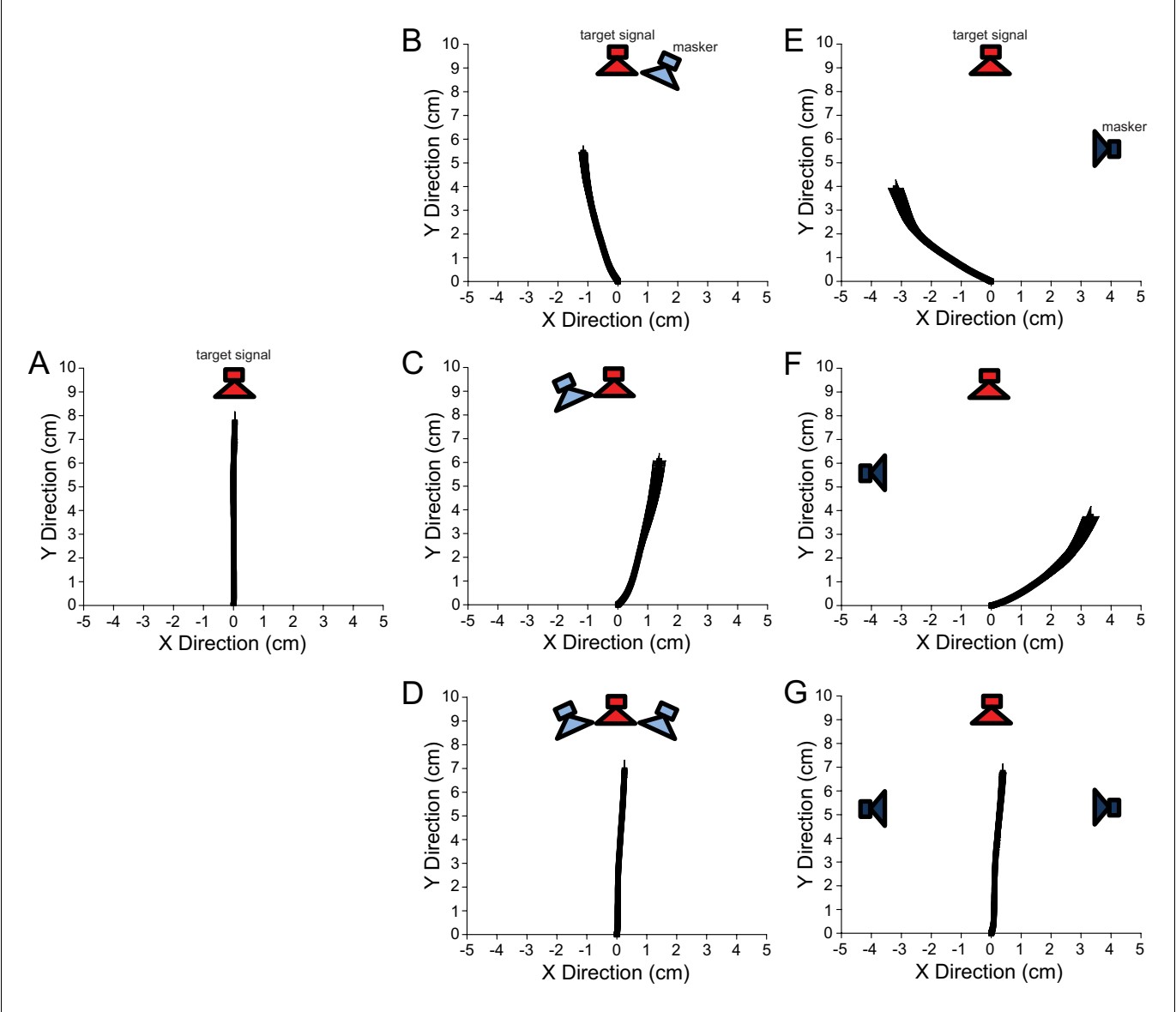

**Figure 5.** Virtual walking trajectories in response to asymmetrical and symmetrical auditory input. Attractive target signal and masker broadcast at 76 dB SPL. Plots represent average walking responses ± SEM to a frontal signal broadcast (A) in isolation, or with (B) a masker in close proximity to the right (6° separation), (C) a masker in close proximity to the left (6° separation), (D) two coherent maskers at ±6° from the signal, (E) a masker to the right with a 90° separation, (F) a masker to the left with a 90° separation, and (G) two coherent maskers at ±90° from the signal. Red speakers indicate the location of the target signal. Light blue and dark blue speakers indicate the location of the masker. A masking source positioned asymmetrically about the midline body axis caused flies to divert their walking responses away from the signal and masker locations. When two identical maskers were used to balance temporal pattern interference on both sides of the auditory system, flies localized the signal location accurately.

The following source data is available for figure 5:

**Source data 1.** Virtual walking path in response to the target signal presented in isolation.

**Source data 2.** Virtual walking path in response to a target signal and masker (on the right) separated by 6° and presented at equal intensity.

**Source data 3.** Virtual walking path in response to a target signal and masker (on the left) separated by 6° and presented at equal intensity.

**Source data 4.** Virtual walking path in response to a target signal and maskers separated by 6° and presented at equal intensity.

**Source data 5.** Virtual walking path in response to a target signal and masker (on the right) separated by 90° and presented at equal intensity.

*Figure 5 continued*

**Source data 6.** Virtual walking path in response to a target signal and masker (on the left) separated by 90° and presented at equal intensity.

**Source data 7.** Virtual walking path in response to target signal and maskers that are separated by 90° and presented at equal intensity.

## Neural measurements

To determine if peripheral neural responses support the hypothesis that the salience of auditory stimuli is determined by the effective amplitude, we made multi-unit recordings from the left and right frontal (auditory) nerve simultaneously in response to the target signal and masker separated by 90° (*Figure 8A*). We applied signal detection theory to determine the masker level at which masking occurred for the frontal target signal masked by an ipsilateral or contralateral masker (*Figure 8B*). In quiet, auditory-evoked impulses were stereotypically timed with the occurrence of the pulse pattern (target signal) (*Figure 8A*). As masker intensity was increased, acoustically evoked impulses occurred at the onset of the masker and continued to occur randomly in time throughout masker presentation. Target signal-driven activity was readily discernible at high SNR, but became obscured by masker-driven activity at lower SNRs (greater masker levels). The SNR at which target-signal driven activity was masked was lower for the masker-contralateral ear compared to the masker-ipsilateral ear (*Figure 8A*). For the masker-contralateral ear, signal detection occurred up to a masker level of $66.75 \pm 0.73$ dB. However, signal detection from the masker-ipsilateral ear was limited to a significantly lower masker level of $59.66 \pm 0.58$ dB (Paired T-Test: n = 10, t = 7.562, p < 0.001) (*Figure 8C*). In other words, target signal detection was possible at a much higher masker level for the masker-contralateral ear compared to the masker-ipsilateral ear. Thus, the masker-contralateral ear was most sensitive to target signal detection.

## Discussion

*Ormia ochracea* responded with greater walking distances, consistent response latencies, and localized the target signal with improved accuracy in the absence of a masker. Broadcasting the masker over the target signal increased response thresholds and compromised the ability of flies to resolve signal location at all SNRs. Changes in SNR altered walking distance, and to a lesser extent, walking direction. However, increasing the spatial separation from 6° (small separation) to 90° (large separation) between the target signal and masker did not improve signal localization or decrease behavioural response thresholds, but caused flies to divert their walking responses even further away from both sources. More interestingly, the direction of diverted walking was not random, but depended on the masker location.

Beyond demonstrating the potential lack of SRM in *O. ochracea*, our goals were to show how spatial separation between a target signal and masker affects localization responses, and peripheral sensory mechanisms that may account for localization behaviour. In experiments that test for SRM, a masker can be presented from a forward location while a target signal is displaced relative to the masker (*Caldwell et al., 2016*; *Holt and Schusterman, 2007*). Similar to some previous studies that have successfully shown SRM, we chose to present a target signal from a fixed forward direction while a masker was displaced relative to the target signal (*Dent et al., 2009*; *Litovsky, 2005*; *Turnbull, 1994*). This target signal location allowed us to easily assess the accuracy of sound localization (i.e. localization to 0° azimuth in response to a forward target signal). Several response characteristics of open-loop walking phonotaxis in *O. ochracea* would render assessment of localization accuracy difficult when a target signal is displaced relative to a forward masker. Under open-loop tethered walking phonotaxis, *O. ochracea* receive no corrective sensory feedback when orienting on the treadmill. As a result, turns toward a lateral target signal location tend to overshoot actual source azimuth (*Mason et al., 2005*). Furthermore, *O. ochracea* exhibit graded turning responses that vary with sound direction, but saturate for sound presentation angles greater than ±30° azimuth (*Mason et al., 2001*). Both of these considerations would make it problematic to quantify the accuracy of phonotaxis, since the predicted orientation is only clearly defined for a forward-located source.

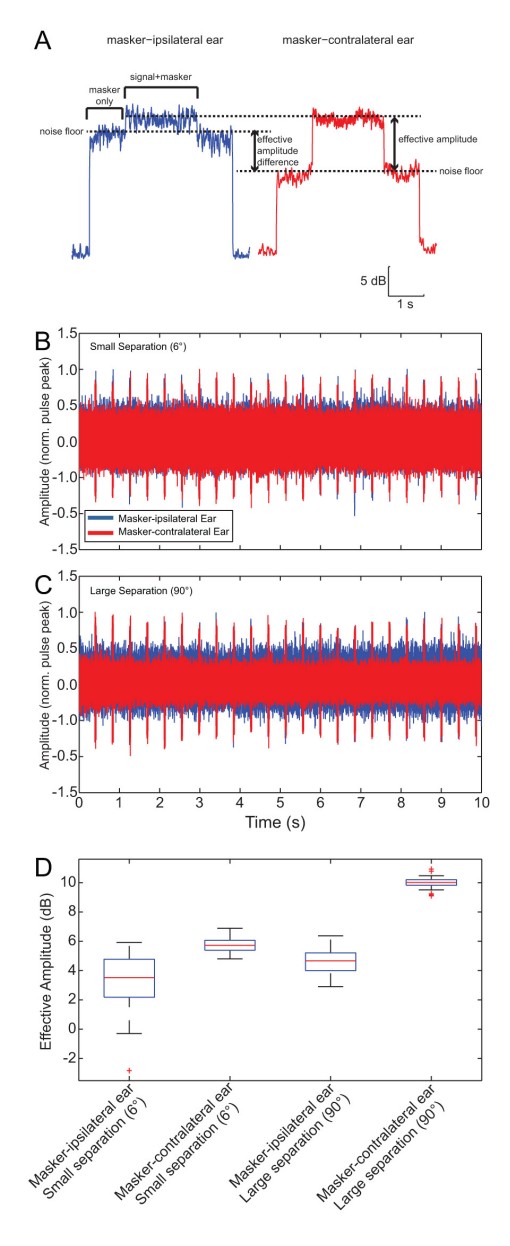

**Figure 6.** Effects of spatial separation on tympanal vibration. (**A**) Exemplar vibration measurements to illustrate the effective amplitude, which is the signal driven response that is above the masker-driven response. Measurement (smoothed with a sliding RMS window) from the masker-ipsilateral ear (blue) and the masker-contralateral ear (red) in response to a signal and masker presented at −6 dB SNR (frontal signal: 70 dB SPL, lateral masker: 76 dB SPL) and separated by 90°. The masker-contralateral ear exhibits a greater effective response amplitude. (**B** and **C**) Tympanal vibration responses to repeated presentations of (forward-located) 2-pulse signal and masker with 6° (**B**) or 90° (**C**) spatial separation. (**D**) Effective response amplitudes vary with noise location. Effective amplitudes are slightly larger in the masker-
*Figure 6 continued on next page*

Diverted walking responses appear to be driven by directional response properties of the two tympana that altered the perceived location of the target signal (*Lee et al., 2009*). For a frontal sound source, both left and right tympana are expected to respond equally in timing and amplitude (*Robert et al., 1996b*), resulting in equivalent afferent activity on both sides of the auditory system. Under this symmetrical stimulation, flies perceived the target signal location to be straight ahead and responded by phonotaxis to a forward location (*Figure 2C*). Our laser vibrometry measurements indicate that adding a masker positioned off the midline axis caused the two tympana to vibrate asymmetrically (*Figures 6* and *7*). The masker-ipsilateral tympanum exhibited a stronger masker-driven response compared to the masker-contralateral tympanum (*Figure 7A*). Consequently, the masker-contralateral tympanum responded with a lower noise floor that allowed for a greater signal-driven effective amplitude of vibration relative to the masker-ipsilateral tympanum. Neural measurements from the left and right auditory nerve also revealed that target signal detection was possible at higher masker levels for the masker-contralateral ear and thus, resulted in better signal detection (*Figure 8*). Consistent with our hypothesis, flies orient to a sound-source direction that is a function of the relative effective amplitude between the two tympana and in the direction of the ear with the best signal detection (*Figure 9*). When a second identical masker was broadcast to equalize masking interference on both sides of the auditory system, symmetry in target signal temporal pattern input was restored and diverted walking responses were corrected so that flies localized the target signal location accurately (*Figure 5*).

The nature of auditory coding in *O. ochracea* suggests that an effect of continuous noise would be to decrease the effective (or perceived) intensity of an attractive stimulus. At the neural level most auditory receptors in *O. ochracea* are Type I afferents that are phasic and precisely mark the onset of a sound pulse with a single spike followed by a refractory period of approximately 4 ms (*Oshinsky and Hoy, 2002*). Type II afferents respond similarly but with several spikes at pulse onset (*Oshinsky and Hoy, 2002*). This is a coding strategy that exploits the predictable, pulsatile pattern of cricket songs by registering the onset timing of successive pulses within a chirp with synchronous bursts of phasic afferent activity. A smaller population of afferents, Type III, are tonic and maintain spiking activity throughout the

*Figure 6 continued*

contralateral ear compared to the masker-ipsilateral ear when there is a 6° spatial separation between the target signal and masker, and this interaural difference is greater for a 90° spatial separation between the target signal and masker.

The following source data is available for figure 6:

**Source data 1.** Raw traces of tympanal vibration in response to combined signal and masker under conditions of masker source located near (separated by 6°) or far (separated by 90°) from signal source.

**Source data 2.** Normalized-amplitude tympanal vibration responses.

**Source data 3.** Smoothed tympanal vibration responses.

**Source data 4.** Effective amplitude measurements.

duration of sound pulses (*Oshinsky and Hoy, 2002*). Continuous noise is not expected to elicit regular activity in Type I or II afferents (although stochastic amplitude fluctuations in the noise envelope may elicit synchronous afferent spikes). The overall effect, therefore, is to set the noise floor above which a signal-driven increment in vibration amplitude is required to elicit a neural response. For a point-source noise, these effects are greater in the noise-ipsilateral ear and will tend to simulate a directional bias of the song location in the direction of the noise-contralateral ear.

At the onset of the masker, our noise stimulus may have evoked Type 1 afferents to fire similarly to the onset of sound pulses within a cricket song. Thus at the onset of the masker, flies oriented transiently to the masker (*Figure 4*) but did not sustain walking (*Figure 3*) in the absence of an attractive temporal pattern. When flies perceived an attractive source over the masker, the strength of phonotaxis increased in a SNR-dependent manner (*Figures 3* and *5*).

In our experiments, band-limited noise was used to interfere with temporal pattern input. In similar experiments, *Wendler (1989)* and *Stabel et al. (1989)* presented a constant tone from an azimuthal location that asymmetrically disrupted temporal pattern input to the cricket auditory system. This resulted in diverted walking responses away from the tone location (*Stabel et al., 1989*; *Wendler, 1989*). Neural measurements revealed that the activity of the tone-contralateral auditory interneurons (AN1 and AN2) better represented the song temporal pattern (*Stabel et al., 1989*). More recent behavioural experiments have shown that song pattern recognition modulates the strength of phonotaxis, while the direction of localization depends on reactive steering to individual sound pulses (*Hedwig and Poulet, 2004*, *2005*; *Poulet and Hedwig, 2005*). These results suggest that crickets responded to a direction that should balance auditory input. When we introduced a second coherent masker on the opposite side of the auditory system to balance temporal pattern interference, symmetrical auditory input was restored and flies accurately localized the frontal attractive sound source. Our results extend these previous findings to show that the mechanically coupled ears of *O. ochracea* function as a sensitive bilateral symmetry detector for song temporal pattern. Flies engage in turning responses in attempt to balance temporal pattern input for accurate sound localization.

Animals often experience lower masked thresholds when masking noise is spatially separated from signals of interest (*Caird et al., 1989*; *Dent et al., 1997*; *Hine et al., 1994*; *Holt and Schusterman, 2007*; *Ison and Agrawal, 1998*; *Schmidt and Römer, 2011*; *Schwartz and Gerhardt, 1989*; *Warnecke et al., 2014*). SRM necessitates that peripheral directionality will give rise to different interaural cues associated with a target signal and noise that vary as a function of spatial separation (*Caldwell et al., 2014*). The magnitude of SRM may be further enhanced by central mechanisms (*Brunnhofer et al., 2016*; *Lin and Feng, 2003*). Despite peripheral directionality provided by the mechanically coupled ears of *O. ochracea*, our behavioural threshold measurements provide no evidence of SRM in *O. ochracea*. Specifically, with increased spatial separations from 6° to 90° azimuth, we did not observe any indication of SRM. This is in stark contrast to some other animals that undoubtedly experience SRM over similar target signal and noise separations (i.e. from 7.5° to 90° separation between a target signal and masker) (*Bee, 2007*). However, our current results do not exclude the possibility that SRM may occur at smaller azimuthal separations (<6° azimuth). Further experiments that measure behavioural response thresholds to a target signal and masker presented from the same spatial location would address this possibility. Even if SRM occurs at smaller spatial separations, our results would suggest that any SRM in *O. ochracea* is rather limited and saturates quickly beyond 6° of spatial separation.

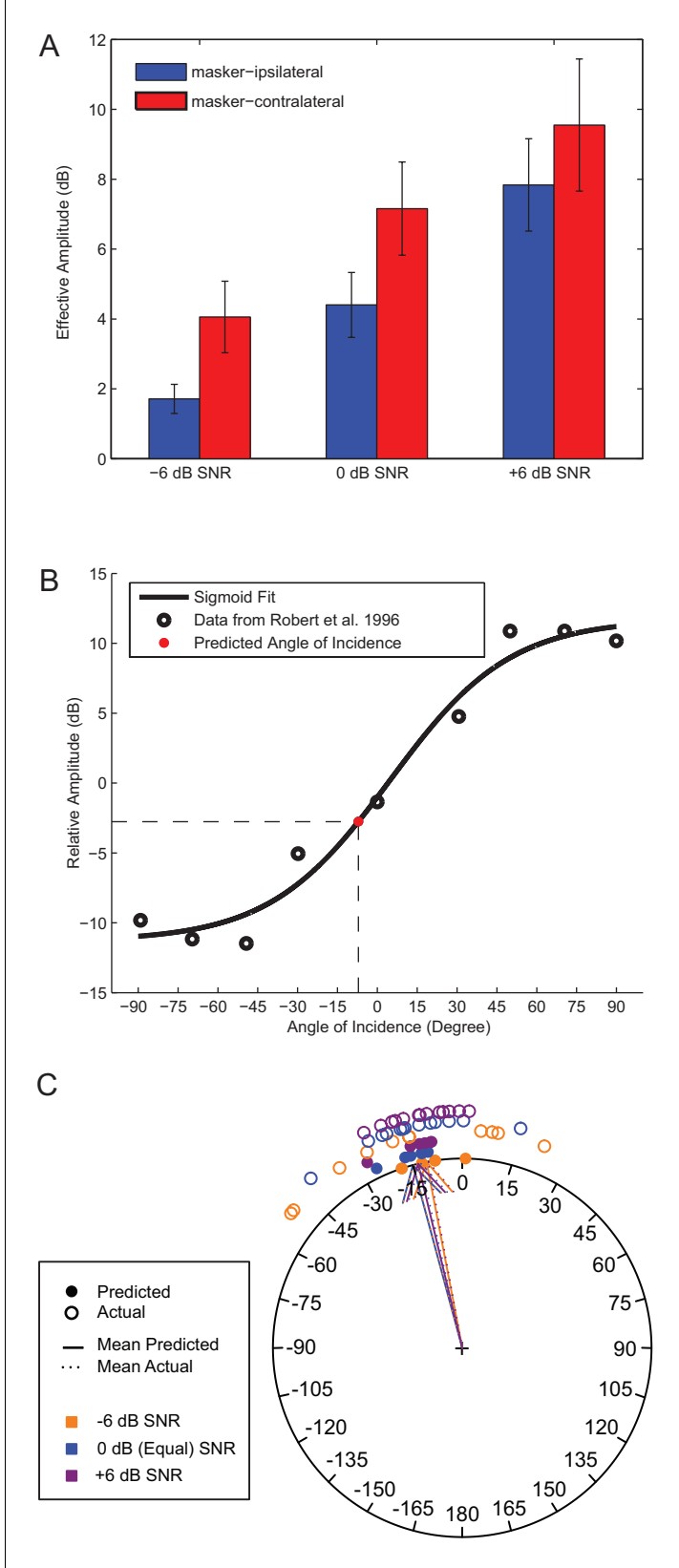

**Figure 7.** Tympanal interaural vibration amplitude difference predict error in sound localization. (**A**) Effective vibration amplitudes measured in LDV experiments to a target signal broadcast simultaneously with an ipsilateral (blue) or contralateral (red) masker at varied SNRs. (**B**) Effective interaural vibration amplitude differences (IVADs)
*Figure 7 continued on next page*

*Figure 7 continued*

were used to generate predictions of the target signal sound direction (red circle) based on previously published directionality measurements (open circles) (redrawn from *Robert et al., 1996b*). (**C**) Mean predicted sound direction (closed circles) generally match with actual behavioural measurements (open circles) across all SNRs (colours). Source data for *Figure 7* is available to download from Dryad Digital Repository under the titles of 'Figure 7 - source data 1' and 'Figure 7 - source data 2' (*Lee and Mason, 2017a*).

We suggest two reasons that may explain this unexpected result. When signal and noise arise from the same spatial location, each ear will experience identical binaural cues (ITD and ILD) from both sources. As signal and noise are spatially separated, differences in these cues associated with each source will occur at both ears. First, the auditory system of *O. ochracea* may be unable to

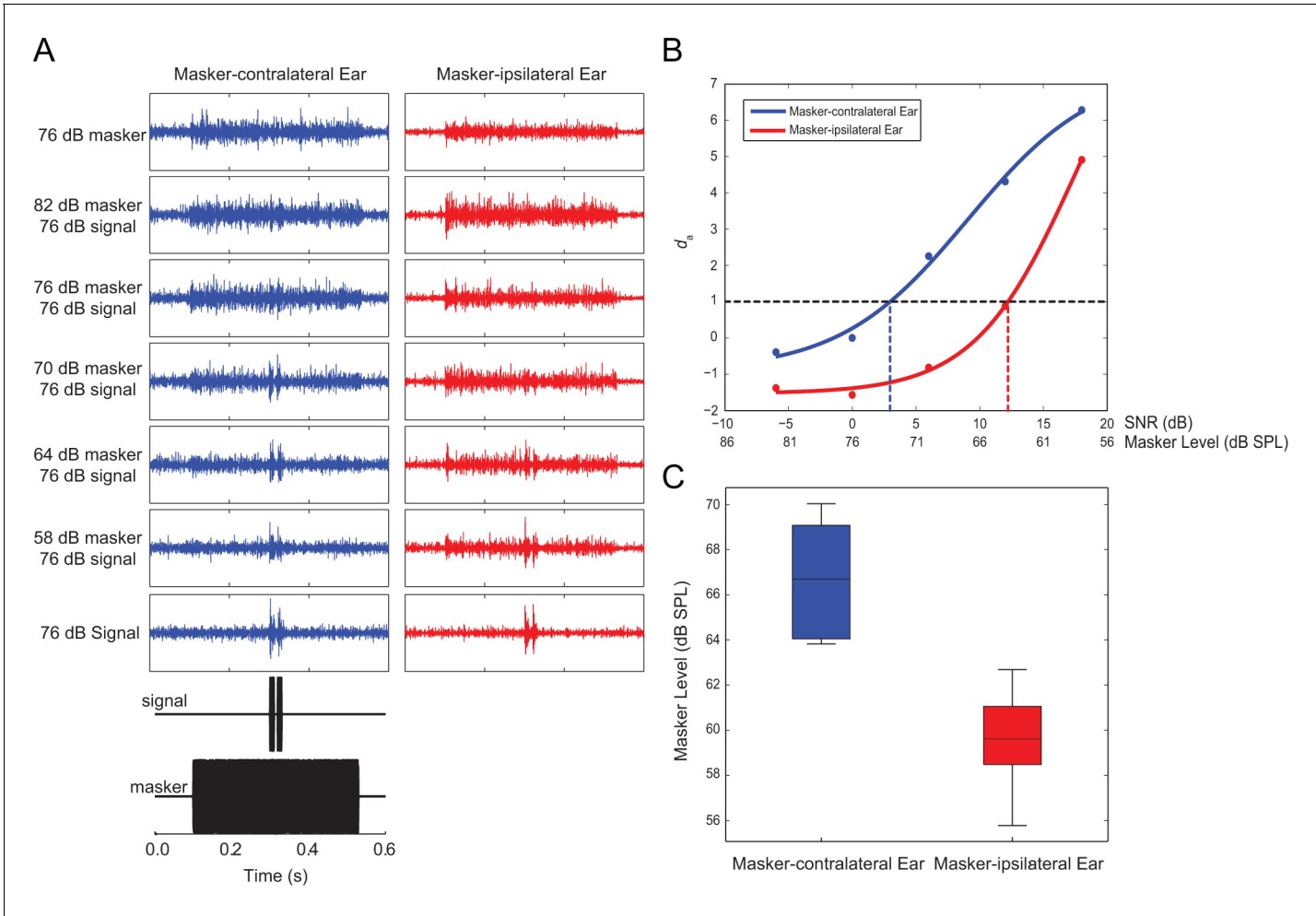

**Figure 8.** Better signal detection in the masker-contralateral ear. (**A**) Exemplar multi-unit recordings from the both sides of the auditory system in response to signal and masker with a 90° spatial separation. Inset (black) shows the time course of the stimulation protocol. The target signal-driven response is apparent at lower SNRs for the masker-contralateral ear (blue) compared to the masker-ipsilateral ear (red). (**B**) Signal detection theory was applied to determine masked thresholds. The mean and variance of impulse rates to the masker and the target signal plus masker was expressed in terms of standard separation ($d_a$) for the masker-contralateral ear (blue) and the masker-ipsilateral ear (red). (**C**) Signal detection was possible at a higher masker level for the masker-contralateral ear (blue) compared to the masker-ipsilateral ear (red).
The following source data is available for figure 8:

**Source data 1.** Root Mean Square (RMS) values calculated from multiunit recordings of left and right auditory nerve.

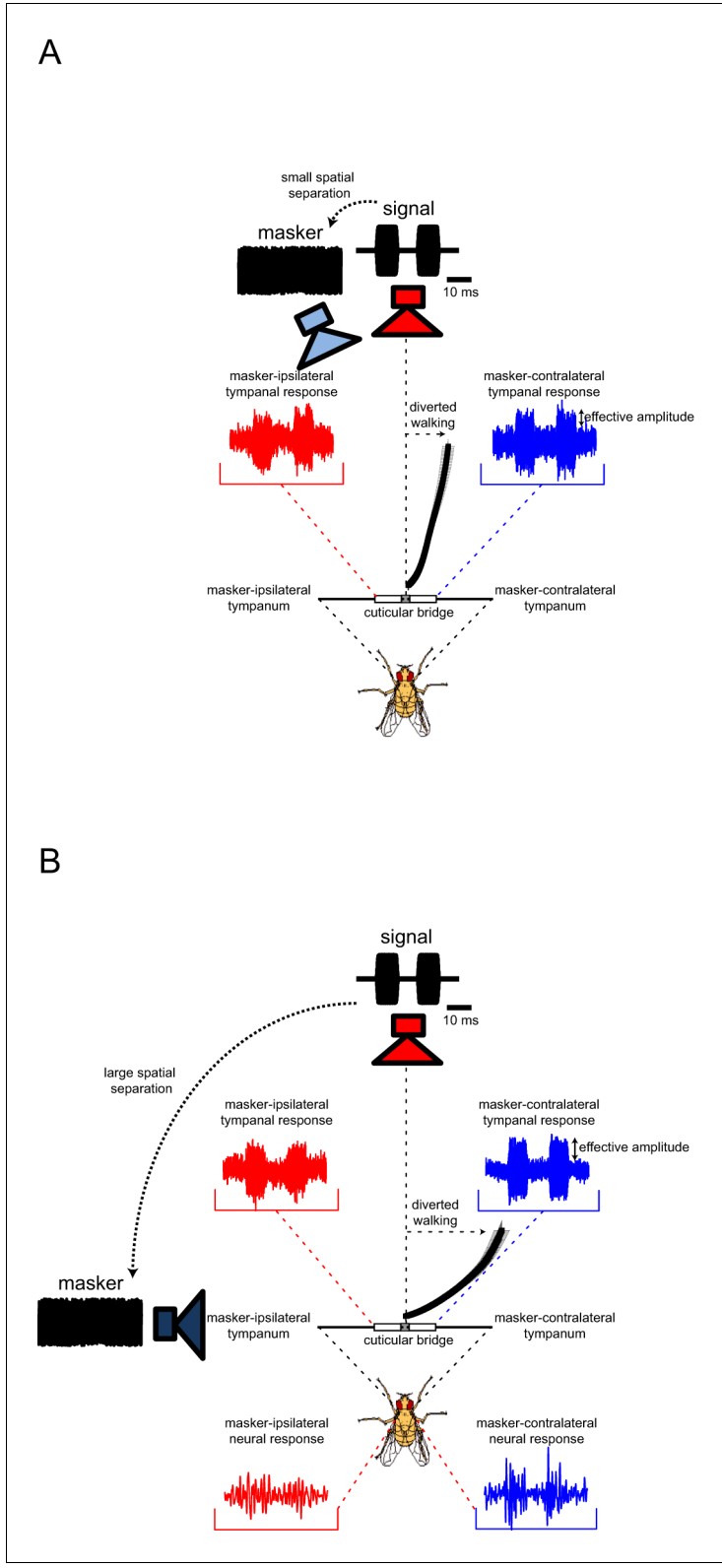

**Figure 9.** Current model of diverted walking phonotaxis in *Ormia ochracea*. (A) 6° and (B) 90° spatial separations between the target signal (cricket song) and masker (band-limited noise). The forward target signal results in equal (symmetrical) stimulation to both tympana while the masker provides greater (asymmetrical) stimulation to the masker-ipsilateral tympana compared to the masker-contralateral tympana (exemplar vibration measurements shown). This results in binaural effective amplitude differences in the detection of the target signal above the

*Figure 9 continued on next page*

*Figure 9 continued*

masker-driven random noise floor. Such binaural effective amplitude differences are larger for larger spatial separations between the target signal and masker. This translates to better signal detection in the masker-contralateral ear (exemplar multi-unit recordings from the auditory nerve shown when the target signal and masker are separated by 90°). Binaural effective amplitude differences potentially encode 'perceived' target signal location. Flies respond by localizing a direction that is diverted away from the target signal and masker.

segregate interaural cues of the target signal from those that are associated with the noise source and this may be one limitation to the function of mechanically coupled ears. SRM does not appear to function in grasshoppers for similar reasons (*Ronacher and Hoffmann, 2003*). In grasshoppers, directional information and temporal recognition are processed in parallel such that input from both sides of the auditory system are summed internally (*von Helversen, 1984*; *von Helversen and von Helversen, 1995*) and thus, binaural cues associated with an attractive signal would be mixed with binaural cues associated with noise. Second, when a point source noise was positioned asymmetrically about the midline axis, flies performed phonotaxis in the direction of the ear with better temporal pattern detection ('better ear'). The noise source provided directional masking that caused a shift in the perceived location of the target signal location away from the actual source location (*Figures 5* and *6*) and flies responded by turning in a direction that would balance temporal pattern input. Under more natural conditions (with freely walking flies), this would still allow successful localization of the target (host) source. Initially, misdirected movement would result in a shift of the relative spatial positions of the target and noise sources allowing flies to arrive at the target by an indirect path. Previous work has shown that for multiple competing attractive sources, flies rely on temporal cues for source segregation, via a precedence effect, combined with a preference for forward locations (*Lee et al., 2009*). We now show that noise can bias the perceived directionality of attractive sources. Future studies will examine how these processes interact to support the surprising spatial hearing abilities of *O. ochracea*.

## Materials and methods

### Animals

Experiments were conducted on lab-reared gravid female *Ormia ochracea* derived from a population originally collected in Gainesville FL. Flies were maintained in environmentally controlled chambers (Power Scientific, Inc. Model DROS52503, Pipersville PA, USA) at 25°C and 75% humidity on a 12 hr:12 hr light:dark regime and fed nectar solution (The Birding Company, Yarmouth MA, USA) *ad libitum*.

### Acoustic stimuli

The attractive stimulus (target signal) was a synthetic song modeled after *Gryllus rubens* trills that was constructed from ramped (1 ms on/off) 10 ms duration sound pulses with a carrier frequency of 5 kHz. In behavioural experiments (Experiments 1, 2, and 3) individual sound pulses were repeated at 50 pulses/s to form 2 s trills. In laser vibrometry experiments, the target signal consists of individual sound pulses repeated at 50 pulses/s to form 2 s trills or a simplified version was generated by shortening the 2 s trill to 40 ms (2-pulse stimulus).

The 2-pulse stimulus was used in neurophysiology experiments. Noise stimuli (masker) were 4 s bursts of band-limited (~2–7 kHz) random noise. Stimulus waveforms (*Figure 1B*) were synthesized in MATLAB (R2009b, MathWorks, Natick, MA) and converted to analog signals using National Instruments acquisition hardware (NI USB-6251, 44100 Hz, Austin TX), amplified (Realistic SA-10 Solid State Amplifier MOD-31-1982B, Taiwan) and broadcast through Skullcandy earbuds (Model: INK'D, China). Sound sources were calibrated at the start of each day of experiments. Target signal levels were software controlled and calibrated (peak RMS, re. 20 μPa) using a probe microphone (B&K Type 4182, Denmark) powered by B&K Nexus Conditioning Amplifier (Denmark). The continuous masker lacked temporal structure and contained broad spectrum energy that overlapped and masked temporal and spectral features of the 5 kHz cricket song (*Figure 1B*). Masker levels were

calibrated as long-term RMS amplitudes of specific SPLs (re. 20 µPa; C- weighted) over 30 s using a B&K sound level meter (Type 2231).

## Behavioural experiments

Prior to behavioural testing, flies were anaesthetized by cooling on ice for 5 min, then mounted to a tether attached to a micromanipulator (Narishige BC-4) with a small amount of wax applied to the dorsum. Recovery from the tethering procedure usually occurred within 5 min. After recovery, flies were positioned on top of the treadmill system and adjusted with the micromanipulator to assume the natural walking posture (*Mason et al., 2001*). Flies were allowed to acclimate for another 20 min before testing.

## Experimental apparatus

Behavioural measurements were made from tethered flies performing walking phonotaxis on a high-resolution treadmill system situated equidistant (Experiments 1 and 3: 20 cm, Experiment 2: 10 cm) from all test speakers (*Figure 1A*) and surrounded by acoustic echo-attenuating foam. The treadmill system consists of a light-weight table tennis ball held afloat by a constant airstream above a modified optical mouse sensor (ADNS 2620, Avago Technologies, USA). Walking responses were transduced as rotations of the treadmill that actuated the optical mouse sensor to record changes in x and y pixel units at a sampling rate of 2160 Hz (*Lott et al., 2007*). Pixel units were calibrated to actual walking distances by measuring displacement of points on the ball from highspeed video footage (Lightning RDT) synchronized to pixel data captured by the treadmill system (Midas, Xcitex). Data collection by the treadmill system was controlled by Stimprog version 5.14b 'treadmill' developed as a MATLAB GUI that interfaced with the National Instruments data acquisition system to ensure synchronous sound presentation and the recording of walking traces. Stimprog is available via GitHub (https://github.com/Ormia/Stimprog.git, (*Lee and Mason, 2017b*); with a copy archived at https://github.com/elifesciences-publications/Stimprog).

In all experiments, the target signal was presented from a frontal speaker located at 0° azimuth relative to the fly's midline axis. In experiments involving a small spatial separation between the target signal and masker, the masker was broadcast from a speaker directly adjacent and to the left or right of the target signal, giving 6° of spatial separation. In experiments involving a large spatial separation, the masker was broadcast from a laterally positioned speaker located at ±90° azimuth.

### Experiment 1: Estimating behavioural response thresholds

Behavioural response thresholds were estimated using an adaptive tracking procedure as described in *Bee and Schwartz (2009)*. We considered translational movements greater than 1 cm to be valid behavioural responses. To estimate behavioral response thresholds in quiet (absence of a masker), the initial target signal level was set to 76 dB SPL and was decreased in 3 dB steps until a non-response was reached. In a final test, the target signal level was increased by half a step size (1.5 dB). If the subject responded to this final test, this signal level was specified as the upper bound (UB) of the estimate. Otherwise, the final signal level was deemed to be the lower bound (LB) of the estimate. The estimated response threshold was calculated as:

Recognition threshold = $10*\log_{10}[(10^{(UB/10)}+10^{(LB/10)})/2]$

In a within-subjects design, this procedure was repeated for signal and masker speakers separated by 6° and 90° angles. The masker intensity was set to 76 dB SPL. Response thresholds were compared in a repeated measures ANOVA.

### Experiment 2: Localizing an attractive signal in the presence of a directional masker

The goal of this experiment was to determine if cricket song localization is impaired in the presence of a masker and whether increased spatial separation between the target signal and masker would ameliorate this impairment. The target signal (see above) presented at 76 dB SPL in the absence of a masker served as the positive control stimulus. The masker was also broadcast at 76 dB SPL in the absence of an attractive signal as a negative control. Test stimuli consisted of a combination of the standard cricket trill (target signal) broadcast at different levels (70, 76, and 82 dB SPL) simultaneously with the masker to achieve three signal-to-noise ratios (−6 dB, 0 dB, +6 dB SNR) in two

different spatial configurations (6° and 90° spatial separations) to form a 2 by 3 factorial design. Stimuli were timed such that the target signal occurred midway through the masker presentation (beginning one second after the onset of the masker playback and finishing one second before the masker offset) (*Figure 1B*). Flies were tested under a repeated measures design such that 3–6 responses were collected for each unique combination of SNR and spatial separation (number of responses differed because response probability decreased at lowest SNR). A test sequence for a fly commenced with three negative controls, a block of half the test stimuli in random order, one mid-sequence control stimulus, the remaining block of test stimuli in random order, and ended with three more positive and negative control stimulus presentations.

## Data analysis

X and Y coordinates from treadmill data traces were collected at 2160 Hz to construct virtual walking trajectories. Walking velocities were separated into two components: (1) steering velocities and (2) forward velocities were calculated as changes in x or y coordinates respectively over time. Angular heading was calculated as in *Mason et al. (2005)*. Responses collected under the same speaker separations but for the reversed position were combined by mirroring responses relative to 0° azimuth. Repeated responses for the same stimulus conditions were averaged within individuals. Unless otherwise specified, data given as mean ± SEM.

Response latencies and walking distances may indicate signal detectability, but we cannot rule out the possibility that they may also be influenced by the motivation to respond. Steering and forward velocities are associated with walking direction. Steering velocities indicate the strength of turning towards or away from a laterally positioned masker. Forward velocities indicate response strength directed to the song location. Angular headings indicate the actual direction of walking. Separate repeated measures ANOVAs were used to assess the dependence of latency, distance, steering and forward velocities, on spatial separation and SNR. Each ANOVA consist of 2 spatial separations (within subjects)×3 SNRs (within subjects). Currently, there is no consensus on statistical models equivalent to a repeated measures ANOVA for circular data. Repeated measurements of angular data across factors were treated as independent groups and a two-factor circular ANOVA (hk test) was used to compare the effects of SNR and spatial separation on angular heading.

If target signal localization is impaired by the presence of a point-source masker, we predicted that flies would exhibit longer response latencies, shorter walking distances, and walking directions (angular heading) that deviate from the target signal location in the presence of a masker. We further predicted that flies would show improvements in these response measures if the target signal and masker are spatially separated and broadcast at greater SNRs.

## Experiment 3: Examining the effects of balanced temporal pattern interference on localization accuracy

The goal of this experiment was to test if symmetry in temporal pattern input may correct deviated localization responses observed in Experiment 2 (*Figure 4*). A frontally located song results in symmetrical temporal pattern input to both sides of the auditory system. Asymmetry in temporal pattern input was achieved by exploiting the inherent directional response characteristic of the ears to a point source masker. We manipulated the degree of asymmetry by varying the spatial separation between a target signal and masker with identical speaker arrangements as in Experiment 2 (referred to as: *Asymmetric with a 6° (small) spatial separation* and *Asymmetric and with a 90° (large) spatial separation*). To induce symmetrical temporal pattern input in the presence of a point source masker, we introduced an identical (coherent) masker positioned at a location that is a direct mirror reflection of the other noise speaker (*Figures 1A* and *5D,G*). In the *Symmetric and small spatial separation* condition, the signal speaker is flanked by two maskers at ±6°, while in the *Symmetric and large separation* condition the maskers are located at ±90° (*Figure 5D,G*). The control condition was identical to that of Experiment 2. The test stimulus consists of the attractive stimulus broadcast at 76 dB SPL along with a single masker broadcast at 76 dB SPL (*Asymmetric* conditions) or two coherent maskers each broadcast at 70 dB SPL to result in an overall level of 76 dB SPL (*Symmetric* conditions). The timing of signal and masker broadcast was kept identical to timing relationships established in Experiment 2. Each fly was tested five times per acoustic condition (type of symmetry and target signal masker separation) in random order, for a total of 30 test responses per fly.

## Data analysis

If symmetrical temporal pattern input to both sides of the auditory system is critical for accurate song localization, we predicted that the addition of a second masker (see above) should re-establish symmetry and correct deviated localization responses. Mid-response angular heading is a direct measure of walking direction, and was extracted from virtual walking trajectories as in Experiment 2. According to our prediction, mid-angular headings to target signal alone should not differ from responses to *Symmetric* acoustic conditions, but should significantly differ from responses to *Asymmetric* acoustic conditions. We applied the *Watson's U test* to compare differences in angular headings under different acoustic conditions.

## Experiments examining peripheral sensory mechanisms

In separate experiments, we explored peripheral sensory mechanisms that may contribute to diverted walking responses at the level of the tympanum and the auditory nerve. Prior to experiments, flies were mounted on a custom holder. In laser vibrometry experiments, the fly's head was removed to provide access to the two tympana. A laser Doppler vibrometer (LDV) (Polytec OFV 3001 controller, OFV 511 sensor head, or PSV-400) was used to record tympanal vibrations. Responses from the masker-ipsilateral and masker-contralateral tympanum were measured by positioning the focused laser beam on each tympanal pit in separate measurements. In neurophysiology experiments, the fly's head was left intact but the wings, dorsum, flight muscles, and gut were removed to provide access to the location at which the frontal (auditory) nerve converges onto the prothoracic ganglion (CNS). Tungsten electrodes (A-M Systems Inc., 5 MΩ) were positioned on each of the left and right frontal nerves using Leica micromanipulators (model # 117777). Multi-unit recordings were made simultaneously from both auditory nerves and amplified using an A-M Systems Microelectrode Amplifier (Model 1800). Analog signals from laser vibrometry and neurophysiology experiments were digitized with National Instruments data acquisition hardware (sampling rate of 44100 Hz) and saved using custom software (Stimprog 5.2) developed in MATLAB.

Target signal and masker speakers were positioned in the large (90°) spatial separation configuration (as above) and presented at varied SNRs. In laser vibrometry experiments, different SNRs were achieved by varying the signal level relative to the masker at 76 dB SPL. We recorded three sweeps in response to each of 3 SNRs (−6, 0, +6 dB) within subjects. In neurophysiology experiments, different SNRs were achieved by varying the masker level relative to a 76 dB SPL target signal. We recorded 20 sweeps per SNR and tested 6 SNRs (−18 to +6 dB in 6 dB intervals) within subjects.

## Data analysis

In laser vibrometry experiments, we calculated a sliding RMS amplitude for the masker alone during a 1 s epoch leading up to the simultaneous presentation of the signal and masker. A sliding RMS amplitude value was also calculated over an equivalent time window during the signal and masker epoch. RMS values were converted to dB to calculate a dB difference between the response of the tympanum during the masker alone epoch compared to the signal and masker epoch. This dB difference represents the magnitude of tympanal response to the target signal that is above the 'noise floor' (*Figure 6A*). These dB differences were computed in response to an ipsilateral and contralateral masker and statistically compared in a repeated measures ANOVA.

Target signal-driven effective interaural vibration amplitude differences (IVADs) were computed using the dB difference values. Based on these IVAD measurements, we predicted the direction of sound localizations by using eardrum directionality measurements from a previously published study. Data points from a plot of IVAD as a function of incident sound direction (*figure 7B* from *Robert et al. (1996b)*) were digitally captured using Meazure 2.0 (written by Baron Roberts, C Thing Software) and fitted to a sigmoid curve using *fminsearch* in MATLAB. The predicted direction of apparent source location was determined as the angle of sound incidence along the fitted curve that corresponded with our calculated IVAD measurements. These predictions were compared to the actual direction of sound localization measured in behavioural experiments.

In neurophysiology experiments, we applied signal detection theory to determine masked thresholds for the target signal embedded in the masker. Thresholds for counting auditory evoked impulses were determined as three standard deviations above the RMS over an epoch of the recording in the absence of any acoustic stimulation. The mean and variance of impulse rates were

determined over an epoch of masker alone and a separate epoch of target signal plus masker during an equivalent time window of 40 ms. We calculated detection thresholds using $d_a$ (standard separation), a modified version of the more well-known d' (detectability index) because $d_a$ does not make any assumptions of equal variances in the distribution of impulse rates in response to masker alone and target signal in noise. We calculated the standard separation $d_a$ as (*Simpson and Fitter, 1973*):

$$d_a = (\mu_{(n+s)} - \mu_{(n)})/\sqrt{(\sigma^2_{(n+s)} - \sigma^2_{(n)})/2}$$

Sigmoid curves were fitted to $d_a$ values for each masker location using *fminsearch*. The signal detection threshold was determined as the lowest signal level along the fitted curve that corresponded to a $d_a$ value exceeding 1.0.

# Acknowledgements

We thank Drs. Catherine Carr, Andrew King, Andrei Kozlov, and Ole Næsbye Larsen for reviewing our manuscript and providing helpful feedback that resulted in significant improvements to this manuscript. N Lee would like to thank Mark A. Bee for his generous support and feedback throughout the final phase of this project. We thank TJ Walker for assistance in fly collection, D Koucoulas for meticulous care in fly husbandry duties, N Mhatre for assistance with laser vibrometry, MCB Andrade, J Peever, DR Howard, CL Hall, PA De Luca, PA Guerra, S Sivalinghem for helpful discussion, GS Pollack and HE Farris for comments on an earlier version of this manuscript. N Lee would also like to thank MJ Kim for endless support throughout the project.

This work was supported by the Natural Sciences and Engineering Research Council of Canada Discovery Grant to AC Mason, PGS D3 to N Lee; Ontario Graduate Scholarship to N Lee; and Society for Integrative and Comparative Biology grants-in-aid of research to N Lee.

# Additional information

### Funding

| Funder | Grant reference number | Author |
| --- | --- | --- |
| University of Toronto | Ontario Graduate Scholarship | Norman Lee |
| Society for Integrative and Comparative Biology grants-in-aid of research | | Norman Lee |
| Animal Behavior Society | Student Grant | Norman Lee |
| Natural Sciences and Engineering Research Council of Canada | PGS D3 | Norman Lee |
| Natural Sciences and Engineering Research Council of Canada | Discovery Grant | Andrew C Mason |

The funders had no role in study design, data collection and interpretation, or the decision to submit the work for publication.

### Author contributions

NL, Conceptualization, Data curation, Software, Formal analysis, Validation, Investigation, Visualization, Methodology, Writing—original draft, Writing—review and editing; ACM, Conceptualization, Resources, Data curation, Software, Formal analysis, Supervision, Funding acquisition, Validation, Investigation, Visualization, Methodology, Writing—original draft

### Author ORCIDs

Norman Lee, http://orcid.org/0000-0001-6198-710X
Andrew C Mason, http://orcid.org/0000-0001-7719-8500

# Additional files

## Major datasets

The following dataset was generated:

| Author(s) | Year | Dataset title | Dataset URL | Database, license, and accessibility information |
|---|---|---|---|---|
| Lee N, Mason AC | 2016 | Data from: How Spatial Release from Masking May Fail to Function in a Highly Directional Auditory System | http://dx.doi.org/10.5061/dryad.n1h4n | Available at Dryad Digital Repository under a CC0 Public Domain Dedication |

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
