## [Decision Letter]

Thank you for submitting your article "How Spatial Release from Masking May Fail to Function in a Highly Directional Auditory System" for consideration by *eLife*. Your article has been reviewed by three peer reviewers, one of whom, Catherine Carr (Reviewer#1), is a member of our Board of Reviewing Editors, and the evaluation has been overseen by Andrew King as the Senior Editor. The following individuals involved in review of your submission have agreed to reveal their identity: Andrei Kozlov (Reviewer #2) and Ole Næsbye Larsen (Reviewer #3).

The reviewers have discussed the reviews with one another and the Reviewing Editor has drafted this decision to help you prepare a revised submission.

Summary:

Typically, when two sounds are separated in space, they can be more readily differentiated than when they are close. This spatial release from masking is surprisingly not present in the parasitic fly, *Ormia ochracea*, although it has excellent directional hearing. The authors used behavioral measures, modeling, laser vibrometry and extracellular recordings to show that this deficit is a function of the mechanically coupled ears. Their key behavioral result used phonotaxis to show that increasing the spatial separation between the target and masker did not decrease behavioural response thresholds, or the time an animal takes to decide on target location (or its heading). In their experiments, maskers next to the target and at 90° had the same effect on the fly's heading on the trackball (diverted away from the signal and masker locations).

Essential revisions:

One of the reviewers would like to see laser responses for maskers at the two locations (close and far). They also observe that improved physiological analyses would strengthen the argument about which response types contribute to the multiunit response. Another reviewer raises the potentially serious question of why the locations of the signal speaker and noise source were selected. Further details may be found in the reviewer comments below; this point should be addressed with care. Many other useful suggestions were made.

1) Some insights into the fly's choice of heading were obtained by Laser Doppler Vibrometry measurements of tympanal vibrations; the ear contralateral to the stimulus had a larger response. One issue with the laser data is that the authors do not show laser responses when the maskers are at the two locations (close and far). These data should be shown.

2) The ear's response reflects tympanic amplitude, since recordings from where the auditory nerve converges onto the prothoracic ganglion show stronger responses from the nerve contralateral to the source. Although it's understood that physiological recordings from *Ormia* are difficult, the neural data shouldn't be used for raster plots unless the authors can show adequate isolation of single units. Other types of physiological analyses are better for multiunit data.

3) The speculation in the discussion about which response types contribute to the multiunit response could be improved; most auditory receptors are reported to be Type I, firing one spike at stimulus onset, yet these recordings show continuous responses. There are many reasons why this should be so.

4) I am confused by the choice of the speakers' positions. First of all, I do not understand why the authors did not broadcast both the signal and the noise from the same speaker as a control? Since these flies can localize with a precision of about 2°, is it possible that 6° is already sufficiently far apart? What if the spatial release from masking happens partially (or fully) at less than 6^°^?

5) More generally and more importantly, I do not understand why the authors did not position a noise source at 0° (straight ahead) and use a range of positions for the signal speaker? When both the signal and the noise are played from the same location, all auditory input would be equal for the two ears-and the fly would walk straight ahead when the behavioural threshold SNR is reached. Then the measurements can be repeated as the signal speaker is displaced away from the fixed noise speaker (which is always located straight ahead). A change in the SNR threshold for the fly movement towards the signal speaker at a new position would reveal the presence or absence of the spatial release from masking. In addition, when only noise is present straight ahead, the distribution of movement vectors away from the noise source should be symmetrical about the zero-degree axis (i.e., the same probability to walk to the right or to the left). Introducing the signal at different positions and varying its relative strength (loudness) would skew this vector distribution towards the signal speaker, providing another quantitative measure of the spatial release from masking.

6) How is this experimental design different from the one used in this study? It is different because the noise level would always be identical for the two ears, and the signal level in the ipsi- and contralateral ears would vary as a function of the signal location. In contrast, in the authors' design, the signal level is always the same across the two ears while the noise level varies. Hence, the authors' experiments test how the flies walk away from a noise source as a function of its relative strength and location (in the absence and in the presence of the spatially fixed signal), whereas the design suggested here would test how they walk towards a signal source as a function of its relative strength and location (in the absence and in the presence of the spatially fixed noise). These are different variables.

Indeed, imagine two separate neural systems: one controlling the positive phonotaxis towards the signal (the cricket song), the other controlling the negative phonotaxis away from the noise. By changing the relative level of the signal across the two ears one modulates the former, whereas by changing the relative level of the noise across the two ears, one modulates the latter.

Perhaps I am missing something, but it seems to me that in order to quantify the positive phonotaxis it is necessary to vary the location of the signal speaker while using a fixed noise source, rather than the other way around.

[Editors' note: further revisions were requested prior to acceptance, as described below.]

Thank you for resubmitting your work entitled "How Spatial Release from Masking May Fail to Function in a Highly Directional Auditory System" for further consideration at *eLife*. Your revised article has been favorably evaluated by Andrew King (Senior editor), a Reviewing editor, and two reviewers.

The manuscript has been improved but there are some remaining issues that need to be addressed before a final decision about acceptance can be made, as outlined below. In particular, Reviewer 2 would prefer that you do the suggested experiment, and test what happens when the signal and noise arise from the same location. They provide some suggestions if you cannot do this.

Reviewer #2:

"Despite the peripheral directionality provided by the mechanically coupled ears of *O. ochracea*, our behavioural results indicate that *O. ochracea* does not experience SRM. We suggest two reasons that may explain this unexpected result. When signal and noise arise from the same spatial location, […]"

I do not think that the authors can make this conclusion because they did not test what happens when the signal and noise arise from the same location. Indeed, the authors write in their reply to one of my questions: "It is possible that masked thresholds will be greater for a co-localized signal and masker and some release from masking may occur when the signal and masker is spatially separated by 6° azimuth- our data do not eliminate this possibility."

The following claim is therefore also not justified. "First, the auditory system of *O. ochracea* may be unable to segregate interaural cues of the target signal from those that are associated with the noise source and this may be one limitation to the function of mechanically coupled ears." Having read this paper, I still don't know whether the fly does not have SRM or whether it has a super-acute SRM, matching the 2° superacuity of its spatial hearing. I therefore recommend that the authors either specify in the discussion the range of applicability of their claims, for example: our behavioural results indicate that *O. ochracea* does not experience SRM in the range 6-90° or they do the experiment with the signal and noise broadcasted from the same location (0°) as I had suggested. This experiment does not seem technically more difficult than the experiments they already did, and it would, in my opinion, strengthen the paper by demonstrating conclusively whether these files do or do not have SRM.

The authors' reply to my question about the relative locations of the signal and noise seems reasonable, but why are these arguments (the text following "This is an interesting issue.") and the three listed practical considerations not included in the manuscript? Readers would have to read both the article, and the reviews and the authors' replies to have access to this information. It would be better if the article contained it, including the four supporting references, none of which is included in the manuscript.

The authors' reply to my last question (about the two systems) does not make sense to me. The authors write: "More importantly, the direction of these transient walking responses to noise onset is towards the speaker, reinforcing the conclusion that the noise effect in our experiments is a modulation of the perceived direction of the positive phonotactic response, rather than a separate negative response." It does not make sense to me because the authors themselves write that the flies orient transiently to any sound, signal or noise. Within the first 150 ms, the flies simply can't tell the difference. Therefore, the flies' responses within the first 150 ms when they can not tell the difference between the signal and the noise does not reinforce any interpretation of the flies' responses after the 150 ms when they can tell the difference.

It is, however, an interesting and useful paper and I hope it is published after the revisions are made.

Reviewer #3:

The authors carefully answered all questions and greatly improved the manuscript. The message is clear and based on solid experiments. The manuscript is a pleasure to read.

---

## [Author Response]

*Essential revisions:*

*One of the reviewers would like to see laser responses for maskers at the two locations (close and far). They also observe that improved physiological analyses would strengthen the argument about which response types contribute to the multiunit response. Another reviewer raises the potentially serious question of why the locations of the signal speaker and noise source were selected. Further details may be found in the reviewer comments below; this point should be addressed with care. Many other useful suggestions were made.*

*1) Some insights into the fly's choice of heading were obtained by Laser Doppler Vibrometry measurements of tympanal vibrations; the ear contralateral to the stimulus had a larger response. One issue with the laser data is that the authors do not show laser responses when the maskers are at the two locations (close and far). These data should be shown.*

We have added these data (Figure 6).

*2) The ear's response reflects tympanic amplitude, since recordings from where the auditory nerve converges onto the prothoracic ganglion show stronger responses from the nerve contralateral to the source. Although it's understood that physiological recordings from Ormia are difficult, the neural data shouldn't be used for raster plots unless the authors can show adequate isolation of single units. Other types of physiological analyses are better for multiunit data.*

Our rationale for this unorthodox application of raster analysis was as follows. Previous work has indicated that *Ormia* auditory responses are characterized by synchronous bursts of receptor action potentials with small interaural latency differences encoding sound direction. Thus peaks in multi-unit recordings, representing bursts of synchronous receptor firing, may be treated as salient neural events, with the temporal patterning of bursts reflecting the quality of signal copying in the afferent response. We were attracted to this approach because it illustrates the signal detection question clearly and intuitively. We now recognize that because rasters are usually used to represent single-unit spikes, the original figure is potentially misleading. Multiunit data has therefore been re-analyzed and presented in Figure 8. In Figure 8, we now show averaged multiunit recordings made simultaneously from the left and right side of the auditory system. Multiunit responses were converted to RMS values and signal detection theory was applied to the RMS values to determine neural detection thresholds. Our new analysis reveal almost identical results and does not change the interpretation of the data.

*3) The speculation in the discussion about which response types contribute to the multiunit response could be improved; most auditory receptors are reported to be Type I, firing one spike at stimulus onset, yet these recordings show continuous responses. There are many reasons why this should be so.*

This section of the Discussion has been rewritten.

*4) I am confused by the choice of the speakers' positions. First of all, I do not understand why the authors did not broadcast both the signal and the noise from the same speaker as a control? Since these flies can localize with a precision of about 2*°*, is it possible that 6*° *is already sufficiently far apart? What if the spatial release from masking happens partially (or fully) at less than 6*°*?*

It is possible that masked thresholds will be greater for a co-localized signal and masker and some release from masking may occur when the signal and masker is spatially separated by 6° azimuth- our data do not eliminate this possibility. Our data show that with greater increases in spatial separation (i.e. from 6 to 90°), response thresholds do not change, so there is no evidence of SRM over this range. There is, however, a pronounced effect on the accuracy of signal localization and this effect is opposite to what would be expected if spatial separation of signal and noise sources were to enhance signal detection.

*5) More generally and more importantly, I do not understand why the authors did not position a noise source at 0° (straight ahead) and use a range of positions for the signal speaker? When both the signal and the noise are played from the same location, all auditory input would be equal for the two ears-and the fly would walk straight ahead when the behavioural threshold SNR is reached. Then the measurements can be repeated as the signal speaker is displaced away from the fixed noise speaker (which is always located straight ahead). A change in the SNR threshold for the fly movement towards the signal speaker at a new position would reveal the presence or absence of the spatial release from masking. In addition, when only noise is present straight ahead, the distribution of movement vectors away from the noise source should be symmetrical about the zero-degree axis (i.e., the same probability to walk to the right or to the left). Introducing the signal at different positions and varying its relative strength (loudness) would skew this vector distribution towards the signal speaker, providing another quantitative measure of the spatial release from masking.*

We answer Points 5 and 6 together (below).

*6) How is this experimental design different from the one used in this study? It is different because the noise level would always be identical for the two ears, and the signal level in the ipsi- and contralateral ears would vary as a function of the signal location. In contrast, in the authors' design, the signal level is always the same across the two ears while the noise level varies. Hence, the authors' experiments test how the flies walk away from a noise source as a function of its relative strength and location (in the absence and in the presence of the spatially fixed signal), whereas the design suggested here would test how they walk towards a signal source as a function of its relative strength and location (in the absence and in the presence of the spatially fixed noise). These are different variables.*

*Indeed, imagine two separate neural systems: one controlling the positive phonotaxis towards the signal (the cricket song), the other controlling the negative phonotaxis away from the noise. By changing the relative level of the signal across the two ears one modulates the former, whereas by changing the relative level of the noise across the two ears, one modulates the latter.*

Perhaps I am missing something, but it seems to me that in order to quantify the positive phonotaxis it is necessary to vary the location of the signal speaker while using a fixed noise source, rather than the other way around.

This is an interesting issue. We used a frontal target signal (at 0º azimuth) and displaced a masker relative to the target to test for spatial release from masking (SRM). We acknowledge that this approach is only one of several spatial arrangements that may be used to test for SRM. However, we followed an approach that has proven to be effective in studying SRM in other animals (humans: Litovsky, 2005, Best et al., 2005, birds: Dent et al., 2009, harbour seal: Turnball, 1994). Beyond measuring response thresholds, we also wanted to establish how spatial separation between a target signal and masker affects localization accuracy and our approach provided a straightforward means of testing the specific expectation that *O. ochracea* should localize straight head in response to a forward attractive source.

Several practical considerations justify using a forward signal and lateral noise.

1) In the open-loop condition, forward is the only “correct” localization that can be specified. Because the flies get no corrective feedback when orienting on the treadmill, their turns toward laterally located sources tend to overshoot the actual source azimuth.

2) Turning responses (and the underlying sensory correlate) saturate at larger stimulus angles (above ~30-40° azimuth).

3) Freely walking flies (closed-loop condition) orient to face a source within ~150 ms of stimulus onset and thereafter walk to the source directly, rather than using a series of corrective turns (Mason et al., 2005).

So under normal conditions, forward (0 azimuth) is the only stable source location for *Ormia* phonotaxis. Using the treadmill, we can “force” them to respond continuously to a lateral stimulus. But points 1 and 2 above will make it difficult to clearly predict the effect of SRM (or its absence).

Our interpretation of the data is that flies respond to cricket song by detecting the onsets of sound pulses within the temporal pattern of the song. Individual pulse onsets are detected based on the increment in the stimulus envelope relative to the noise floor, which generates synchronous (or near-synchronous) activity in the receptor populations of the two ears. Spatially balanced noise will decrease the effective stimulus amplitude equally in both ears (masking). As the target source is moved to more lateral azimuth, there will be no effect on the perceived source direction, as both ears will be equally masked. At larger source angles, however, there may be some conditions where the stimulus would be completely masked in the contralateral ear, but not in the ipsilateral, due to the directional asymmetry of the stimulus. In this condition, for some stimulus amplitudes, the flies could respond with a larger turn angle than they would in the absence of masking noise. This would be a similar effect to what our data show (deviation of walking direction away from noise), but only for a specific range of stimulus angles.

In contrast, spatially biased noise with a forward-located source generates unequal binaural masking, and imposes a “perceived” directionality on the non-directional stimulus. The noise effect is to modulate positive phonotaxis via differential binaural masking of the attractive stimulus. The overall effect is that flies are “deflected” from the accurate path to the signal source by spatially biased noise. We show that this effect is present in the mechanical response of the tympanal membranes (more clearly now, with the addition of tympanal measurements matching the behavioural conditions), and in the responses of peripheral auditory receptors.

The possibility of independent modulation of two neural systems (for positive and negative phonotaxis, respectively), as suggested by the reviewer, must also be considered. We have revised the figures, and Figure 2 now shows fly walking responses to noise and signal alone. Flies typically make a transient turn towards the direction of a sound source at the onset of any sound in their audible frequency range, but only show sustained walking for cricket-like patterns of amplitude modulation. More importantly, the direction of these transient walking responses to noise onset is towards the speaker, reinforcing the conclusion that the noise effect in our experiments is a modulation of the perceived direction of the positive phonotactic response, rather than a separate negative response.

[Editors' note: further revisions were requested prior to acceptance, as described below.]

*Reviewer #2:*

*"Despite the peripheral directionality provided by the mechanically coupled ears of O. ochracea, our behavioural results indicate that O. ochracea does not experience SRM. We suggest two reasons that may explain this unexpected result. When signal and noise arise from the same spatial location, […]"*

*I do not think that the authors can make this conclusion because they did not test what happens when the signal and noise arise from the same location. Indeed, the authors write in their reply to one of my questions: "It is possible that masked thresholds will be greater for a co-localized signal and masker and some release from masking may occur when the signal and masker is spatially separated by 6 deg azimuth- our data do not eliminate this possibility."*

We have now modified the text in parts of the Introduction and Discussion sections to emphasize that our data only shows no benefits of spatial release from masking for a sound source separated by 6° to 90° azimuth.

*The following claim is therefore also not justified. "First, the auditory system of O. ochracea may be unable to segregate interaural cues of the target signal from those that are associated with the noise source and this may be one limitation to the function of mechanically coupled ears."*

We respectfully disagree. Let us suppose for the moment that SRM does occur at spatial separations less than 6°. If this is the case, our data would suggest that the magnitude of release from masking saturates at and beyond 6° of spatial separation and that *O. ochracea* gain no further advantage in having greater spatial separations between the signal and masker. Even if *O. ochracea* experiences SRM, our data shows that *O. ochracea* are still unable to separate cues associated with the target signal from those that are associated with the masker. Localization responses are diverted away from the target signal and masker regardless of the whether there is a small (6°) or large (90°) spatial separation (Figure 4 and Figure 5). In these diverted walking responses, we believe that the masker introduced directional cues that were not successfully segregated from target-driven directional cues.

*Having read this paper, I still don't know whether the fly does not have SRM or whether it has a super-acute SRM, matching the 2*°*superacuity of its spatial hearing. I therefore recommend that the authors either specify in the discussion the range of applicability of their claims, for example: our behavioural results indicate that O. ochracea does not experience SRM in the range 6-90*° *or they do the experiment with the signal and noise broadcasted from the same location (0*°*) as I had suggested. This experiment does not seem technically more difficult than the experiments they already did, and it would, in my opinion, strengthen the paper by demonstrating conclusively whether these files do or do not have SRM.*

We have revised the text in the Introduction and Discussion to qualify the interpretation of our results. Whether *O. ochracea* experiences SRM at smaller target signal and masker separations (<6° spatial separation) is an interesting question that we will plan to address in a follow-up study when flies become available.

*The authors' reply to my question about the relative locations of the signal and noise seems reasonable, but why are these arguments (the text following "This is an interesting issue.") and the three listed practical considerations not included in the manuscript? Readers would have to read both the article, and the reviews and the authors' replies to have access to this information. It would be better if the article contained it, including the four supporting references, none of which is included in the manuscript.*

We have included this text and supporting references in the Discussion.

*The authors' reply to my last question (about the two systems) does not make sense to me. The authors write: "More importantly, the direction of these transient walking responses to noise onset is towards the speaker, reinforcing the conclusion that the noise effect in our experiments is a modulation of the perceived direction of the positive phonotactic response, rather than a separate negative response." It does not make sense to me because the authors themselves write that the flies orient transiently to any sound, signal or noise. Within the first 150 ms, the flies simply can't tell the difference. Therefore, the flies' responses within the first 150 ms when they can not tell the difference between the signal and the noise does not reinforce any interpretation of the flies' responses after the 150 ms when they can tell the difference.*

Point taken. Initial responses to sound onset are indiscriminate and therefore we cannot argue that they are indicative of any sustained response to the noise stimulus. The main intention of this argument, however, was to point out that there is no sustained response to a noise stimulus, only this indiscriminate onset-response. Flies do not walk away from noise by itself.

This being the case (no response to noise alone), then the distinction between our interpretation (directionality of positive response is modulated by interaural differences in noise-floor, and consequently perceived signal amplitude) and the proposed two-system model (separate positive and negative responses to signal and noise) becomes somewhat esoteric, since the negative response to noise is only manifest as a bias in the positive signal-driven response.

In addition, our measurements of peripheral auditory responses show that the interaural difference in effective signal amplitude matches the interaural amplitude difference that would be expected for a source azimuth (in quiet conditions) corresponding to the observed behaviour (angle of phonotaxis). The simplest interpretation for all of this is that the signal, at the noise-ipsilateral ear, is more strongly masked than at the noise-contralateral ear, resulting in an asymmetry of signal input that mimics a directional source location.